# TivTok: Broadcasting Time-Invariant Tokens for Scalable Video Tokenization

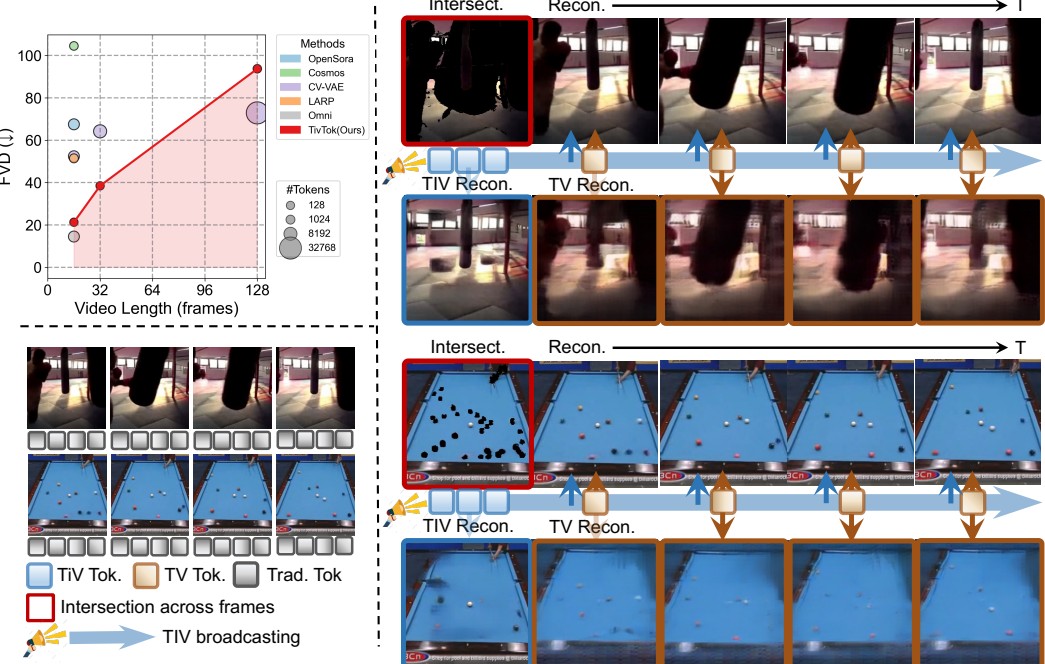

Figure 1: **TivTok** explicitly decouples videos into time-invariant (TIV) tokens and time-variant (TV) tokens, achieving $2.91\times$ higher compression efficiency than traditional tokenizers while maintaining comparable reconstruction quality in long video tokenization by reusing the TIV tokens.

## ABSTRACT

Video tokenization is a critical bottleneck for learned video compression and generation. Existing methods often fail to adapt to the uneven information density of videos, underutilize temporal redundancy, and overlook the reusability of shared content. We present **TivTok** (*Time-Invariant Tokenizer*), a transformer-based tokenizer that explicitly decouples videos into **time-invariant (TIV) tokens**, which capture global information shared across frames, and **time-variant (TV) tokens**, which encode frame-specific residual details. The encoder is designed with tailored attention masking to enforce this factorization, enabling the invariant component to capture not only static elements but also temporally coherent patterns such as consistent motion trajectories. In decoding, a broadcast mechanism reuses TIV tokens across frames, reducing complexity from quadratic to linear in video length. We further extend this approach to long videos through cross-chunk reuse, enabling scalable compression. Experiments show that TivTok improves reconstruction quality with FVD of 12.65 in the traditional $16 \times 256 \times 256$ setting and achieves a $2.91\times$ gain in compression efficiency for $128 \times 256 \times 256$ videos compared to state-of-the-art methods.

## 1 INTRODUCTION

Generative models have achieved remarkable success across diverse downstream applications, including visual content generation (Blattmann et al., 2023a; Rombach et al., 2022; Blattmann et al.,

2023b; Liu et al., 2024; Huang et al., 2025b), cinematic production (Huang et al., 2024; Chen et al., 2024b; Huang et al., 2025a), and industrial simulation (Zheng et al., 2024a; Ren et al., 2024a;b; Chen et al., 2025b; Agarwal et al., 2025). Their success is largely driven by the insight that pixel-space visuals is highly redundant; projecting these high-dimensional representations into compact latent spaces significantly reduces computation and shifts focus to semantic structure, enabling sharper and higher-quality generations (Rombach et al., 2022; Blattmann et al., 2023a). However, video tokenization remains challenging, since it must compress data far more aggressively than images to handle the rapid growth in data volume with longer sequences, while handling the large yet intricate temporal redundancy.

Tokenizing videos requires handling the uneven distribution of information across frames. Encoding each frame or chunk independently introduces large amounts of redundancy, since much of the content—such as scene structure, object appearance, and smooth motion—persists over time. An effective tokenizer should capture these shared patterns compactly, while leaving only the unpredictable details to be represented frame by frame. Just as importantly, the representation should be reusable and extendable as the sequence continues, avoiding the need to re-encode what is already known. Recent work in video tokenization has begun to move in this direction, going beyond simple frame-based encoding to capture richer temporal structure.

One direction extends image tokenizers by adding temporal compression layers (Blattmann et al., 2023b), often with 3D convolutions (Li et al., 2024; Agarwal et al., 2025; Zhao et al., 2024; HaCohen et al., 2024), to handle the extra time dimension. Later methods go further by breaking away from frame-based structure altogether: they patchify the video into 1D token sequences and use transformers to compress them into holistic tokens that summarize global information through attention (Yu et al., 2024a; Huang et al., 2025b; Bachmann et al., 2025; Wang et al., 2024a; Yan et al., 2024; Li et al., 2025). This approach moves beyond a fixed $H \times W \times T$ resolution and allows more flexible allocation of representation capacity (Beyer et al., 2025), though its quadratic attention cost makes scaling to long videos difficult.

A second direction tackles temporal redundancy more directly by decomposing videos into a context frame (often an aggregation of all frames) and relative motion with respect to this frame (Tan et al., 2024; Tian et al., 2024b; Yu et al., 2024b; Wang et al., 2025). This strategy allows longer video reconstruction by reusing the stable context, but its explicit decomposition can oversimplify video structure, which often fails when backgrounds or scenes change dramatically.

Driven by the need to both cut redundancy and promote reusability, we propose **TivTok** (*Time-Invariant Tokenizer*), a tokenizer that extracts temporal invariants through Time-Invariant (TIV) tokens and reuses them across frames to improve video compression. As is shown in Figure 1, we decouple videos into two complementary components: time-invariant representations, which capture semantic invariants rather than pixel persistence; and time-variant representations, which encode the residual, frame-specific details. To realize this factorization, we design a transformer-based architecture with masked attention that enforces a clean separation between invariant and variant tokens. By reusing TIV tokens across video chunks, TivTok naturally scales to long videos and achieves superior compression: it improves reconstruction quality with FVD of 12.65 in the standard $16 \times 256 \times 256$ setting and delivers a $2.91\times$ gain in compression efficiency for $128 \times 256 \times 256$ videos compared to existing approaches. We summarize our main contributions as follows:

- We propose a new paradigm for efficient video tokenization that separates and reuses shared time-invariant information across frames while encoding only frame-specific residuals.
- We design a transformer-based framework with tailored attention masking and decoding to control information flow between time-invariant and time-variant components.
- We enable scalable long video tokenization by reusing time-invariant tokens across frames and chunks, reducing tokenization complexity from quadratic to linear in video length.

## 2 RELATED WORK

### 2.1 FROM IMAGE TOKENIZER TO VIDEO TOKENIZER

Following the tremendous success of the encode-generate paradigm in image generation (Rombach et al., 2022), researchers have developed video tokenizers by extending the dimensionality of exist-

ing image tokenization methods. The core idea underlying these approaches is to treat video tokenization as a natural extension of image compression by adding temporal dimensions to spatial processing architectures. These dimension-extended methods can be categorized into two main lines of works: **Downsample-based Video Tokenizers.** Early work (Blattmann et al., 2023b) first explores adapting image tokenizers for video tokenization by encoding videos frame-by-frame. Subsequent methods (Zhao et al., 2024; Agarwal et al., 2025; Chen et al., 2024a; Tang et al., 2024) extended 2D convolutions to 3D convolutions for temporal downsampling, achieving higher compression ratios while proposing various optimization techniques to facilitate training. CV-VAE (Zhao et al., 2024) leverages 2D convolutions pretrained on images to regularize video tokenizers, improving training efficiency. VidTok (Tang et al., 2024) incorporates multiple techniques including FSQ to improve codebook utilization and compression efficiency. Cosmos (Agarwal et al., 2025) employs 3D Haar wavelets to enhance model performance. **Holistic Tokenizers.** TiTok (Yu et al., 2024a) pioneered the use of transformer architectures to compress images into 1D learnable tokens, enabling higher compression rates through global receptive fields. This approach has inspired subsequent works exploring 1D tokenization for images (Huang et al., 2025b; Tian et al., 2024a) and videos (Wang et al., 2024a; Yan et al., 2024; Li et al., 2025). However, directly applying such methods to videos encounters significant challenges, as video patches vastly outnumber image patches, leading to quadratic computational growth and increased learning complexity that hinders effective video compression. Despite various attempts to explore such compression strategies, these methods remain limited to low-resolution video compression. Different from these dimension-extension approaches, we focus on the fundamental temporal redundancy characteristics of videos by reusing shared information across consecutive frames rather than simply extending spatial processing, thereby achieving superior compression efficiency.

## 2.2 Decompose-based Video Tokenizer

Traditional video compression (e.g., H.264/MPEG-4 AVC (Richardson, 2004) and AV1 (De Rivaz & Haughton, 2019)) has long recognized the fundamental principle of temporal redundancy exploitation through decomposed encoding strategies. In H.264, for instance, P-frames leverage motion compensation by referencing spatial blocks up to 16×16 pixels from previously encoded frames, encoding only the residual differences between the predicted and actual content. This decomposition strategy effectively eliminates temporal redundancy by avoiding redundant encoding of similar visual content across consecutive frames. Recently, numerous works (Wu et al., 2024; Jin et al., 2024; Yu et al., 2024b) have followed this paradigm to explore more efficient video tokenizers through learned decomposition. CMD (Yu et al., 2024b) first proposes content-motion decomposition, encoding videos into a 2D content frame and low-dimensional motion latents to capture static and dynamic information separately. Reducio (Tian et al., 2024b) employs an image-conditioned decoder while maintaining a reference image to achieve high-quality video reconstruction with reduced storage requirements. SweetTok (Tan et al., 2024) separately encodes the first frame and subsequent residual frames, explicitly modeling temporal dependencies through learned residual representations. HiVAE (Liu et al., 2025) decomposes videos into high-frequency and low-frequency components, enabling specialized compression strategies for different temporal scales. However, these predefined decomposition methods suffer from rigid structures that poorly adapt to diverse video content and lack reusability, thus failing to scale efficiently to longer sequences. Moreover, the predefined decomposition introduces additional optimization complexities. In contrast, we propose extracting and reusing temporal invariants across frames, avoiding redundant encoding. Experiments demonstrate that our temporal invariants easily extend to long video sequences while achieving 2.91× higher compression efficiency.

## 3 Method

### 3.1 Preliminary: Transformer-based Holistic Visual Tokenizer

Pioneered by TiTok (Yu et al., 2024a), transformer-based holistic tokenizers have become a popular choice for visual tokenization. Their key idea is to distill a compact set of 1D global latents from all input patches by leveraging the transformer's global receptive field.

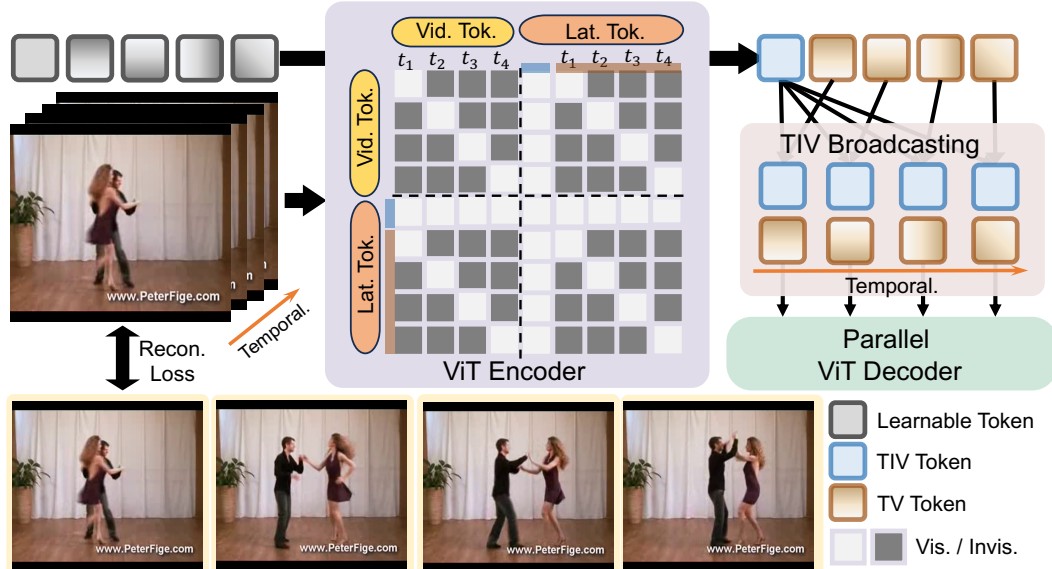

Figure 2: **TivTok** employs a transformer-based tokenizer architecture, consisting of an encoder with dual-range attention masks and a decoder with a TIV Token Broadcasting mechanism, which isolates shared versus frame-specific content and ensures reusability of TIV tokens across all time steps.

Given a video $V \in \mathbb{R}^{3 \times T \times W \times H}$, the tokenizer first *patchifies* $V$ with a fixed downsampling ratio $(f_T, f_W, f_H)$, producing patch features $X \in \mathbb{R}^{d \times \frac{T}{f_T} \times \frac{W}{f_W} \times \frac{H}{f_H}}$. These flattened patches are concatenated with a set of learnable tokens $Z \in \mathbb{R}^{d \times N_z}$ to form $\tilde{Z} = [\text{Flatten}(X); Z]$.

This combined sequence is then passed through a transformer encoder $E(\cdot)$. Through self-attention, the latent tokens absorb global information from all patches across the video. After encoding, the latent tokens are quantized with $Q(\cdot)$ to form a compact representation $\hat{Z}$ that captures the essential content of the video in a discrete code space.

During decoding, learnable patch queries $Q_p$ and the latent codes $\hat{Z}$ are processed by a symmetric transformer decoder $D(\cdot)$ to recover patch features $\hat{X} = D([Q_p; \hat{Z}])$, which are then upsampled to the original resolution. This entire process can be summarized as

$$\hat{Z} = Quant\big(E([\text{patchfy}(V); Z])\big), \qquad \hat{V} = \text{Unpatchify}\big(D([\hat{Z}; Q_p])\big). \tag{1}$$

However, because the number of patches increases linearly with video length, the computational cost of self-attention grows quadratically; both encoding and decoding scale as $O(T^2)$.

## 3.2 DECOUPLING TIME-INVARIANT AND TIME-VARIANT TOKENS

From an information-theoretic perspective, a video sequence can be regarded as a collection of frames with substantial shared content. Its *entropy* provides a natural measure of the information content, and thus reflects the number of tokens (or bits) required for compression.

Consider a video sequence of length $T$. Explicitly capturing the shared component $C$ across frames can substantially reduce sequence entropy. In particular, the reduction can be quantified as

$$H_{\text{indep}} - H_{\text{shared}} \geq \sum_{t=1}^{T} I(X_t; C) - I(C; X_{1:T}) \approx (T-1)I(C; X_{1:T}) \gg 0, \tag{2}$$

where $H_{\text{indep}}$ is the sum of per-frame entropies if frames are encoded independently, and $H_{\text{shared}}$ accounts for the shared information $C$. A detailed derivation is provided in Appendix A. This analysis shows that explicitly capturing and reusing temporal redundancy can dramatically reduce the number of tokens needed to represent a video.

Building on this, we explicitly model the shared *time-invariant* component $C$, which captures redundant information persisting across frames—not only pixel-level persistence, but also *semantic*

*invariants* such as scene geometry, object structure, and consistent visual elements that remain stable throughout the video sequence. Encoding this continuity in the time-invariant component prevents redundant re-encoding of scene-level information in every frame, and leaves the unpredictable, frame-specific residuals to the time-variant component.

To this end, we propose a *decoupled* token representation that factorizes a video into two complementary components:

- **Time-Invariant Tokens (TIV)**, $Z_{TIV} \in \mathbb{R}^{N_{TIV} \times D}$, which encode information shared across frames and can be reused to extend representations to longer videos

- **Time-Variant Tokens (TV)**, $Z_{TV} \in \mathbb{R}^{T \times N_{TV} \times D}$, which preserve frame-specific details unique to each time step.

Formally, for a video $V \in \mathbb{R}^{3 \times T \times H \times W}$, the entire sequence is represented as $[Z_{TIV}, Z_{TV}^{(1)}, Z_{TV}^{(2)}, \ldots, Z_{TV}^{(T)}]$. For an individual frame at time step $t$, its content is compactly described by the token pair $[Z_{TIV}, Z_{TV}^{(t)}]$.

### 3.3 Tokenizer Design for TIV/TV Decoupling

We design a transformer-based tokenizer that explicitly decouples and tokenizes the time-invariant (TIV) and time-variant (TV) components (Figure 2). The encoder isolates shared versus frame-specific content, and the decoder ensures reusability of TIV tokens across all time steps.

To enforce the intended factorization, we constrain the visibility of tokens in the encoder through an attention mask. TIV tokens are granted global visibility: for a video $V = \{X_1, \ldots, X_T\}$, each TIV token attends to all frame patches $\{X_t\}$ as well as all TV tokens. In contrast, each TV token at time step $t$ has only local visibility, restricted to its own frame patches $X_t$, the TIV tokens, and itself.

Formally, the encoder updates are defined as

$$Z'_{\text{TIV}} = \text{Attn}\big(Z_{\text{TIV}}, [Z_{\text{TIV}}, Z_{\text{TV}}^{(1)}, \ldots, Z_{\text{TV}}^{(T)}, X_1, \ldots, X_T]\big),$$
$$Z_{\text{TV}}^{(t)'} = \text{Attn}\big(Z_{\text{TV}}^{(t)}, [Z_{\text{TIV}}, Z_{\text{TV}}^{(t)}, X_t]\big). \tag{3}$$

This dual-range masking is designed to encourage TIV tokens to aggregate shared information across the sequence, while guiding TV tokens to primarily capture frame-local residuals. Using causal masking for TV tokens may seem natural for autoregressive generation, but it would both duplicate information already stored in the TIV tokens and raise the cost to quadratic in $T$. Limiting TVs to single-frame visibility keeps the roles cleanly separated and maintains efficiency, reducing overall encoding complexity from $O\big(T^2 \cdot (N_{\text{TIV}} + N_{\text{TV}})\big)$ to $O\big(T^2 \cdot N_{\text{TIV}} + T \cdot N_{\text{TV}}\big)$.

In the decoder, we propose a *TIV Token Broadcast* mechanism to facilitate reuse of shared information. After encoding, the TIV tokens are broadcast to every time step and recombined with the corresponding TV tokens, so that each frame is decoded as

$$\hat{X}_t = D\big([Z_{\text{TIV}}, Z_{\text{TV}}^{(t)}]\big), \quad t = 1, \ldots, T, \tag{4}$$

where $D(\cdot)$ denotes the transformer decoder. Since all frames reuse the same TIV tokens, they can be decoded in parallel without interfering with one another. This design explicitly reuses shared content and reduces decoding complexity from quadratic in the video length ($O(T^2)$) to linear ($O(T)$), thereby improving efficiency and scalability for long video generation.

## 3.4 BROADCASTING TIV TOKENS FOR LONG VIDEO COMPRESSION

---

**Algorithm 1:** TIV-Broadcast Training Algorithm for Long Video Redundancy

---

**Input:** Long video $\{X_{1:TK}\}$ with $K$ chunks of length $T$, pre-specified distribution $p \in \mathbb{R}^K$;

**Output:** Decoded video $\hat{X}_{1:TK}$;

**1. Parallel Encoding:**

**for** $i = 1, \ldots, K$ **do**

    Encode chunk $X_{1:T}^{(i)} \to Z_c^{(i)}, \{Z_s^{(i,t)}\}_{t=1}^T$

**2. Shared Token Merging:**

Merge TIV tokens: $\bar{Z}_c = \frac{1}{K} \sum_{i=1}^K Z_c^{(i)}$

**3. Token Reorganization:**

$$\mathcal{Z} = [\bar{Z}_c, Z_s^{(1,1)}, \ldots, Z_s^{(1,T)}, \ldots, Z_s^{(K,1)}, \ldots, Z_s^{(K,T)}]$$

**4. Propagation Decoding:**

**for** *each frame t in parallel* **do**

    Broadcast $Z_c^{(k)}$ to frame $t$ and decode with its specific tokens $Z_s^{(i,t)}$ to reconstruct $\hat{X}^{(i,t)}$;

**5. Update:** Compute $\mathcal{L}(\hat{X}, X)$, update parameters

**Complexity:** Encoding and decoding cost scales as $\mathcal{O}(K)$ instead of $\mathcal{O}(K^2)$.

---

Following Eq. 2, we extend our analysis to long videos divided into multiple chunks. For a *traditional tokenizer*, compressing a $K$-chunk video requires a proportional increase in the total number of tokens, and the computational cost grows quadratically with the total length $T$.

In contrast, motivated by our earlier observations, we explore reusing *cross–chunk redundancy* to achieve more efficient video compression. Intuitively, a video's global shared information is relatively stable, so the common content extracted from one chunk can serve as an estimate of the global schema of the entire video. Building on this insight, we propose a training strategy for cross–chunk reuse of shared tokens, as illustrated in Algorithm 1.

Specifically, for a long video $\{X_{1:TK}\}$ composed of $K$ chunks of length $T$, we first encode all $K$ chunks in parallel and merge the temporal invariant tokens by averaging: $\bar{Z}_c = \frac{1}{K} \sum_{i=1}^K Z_c^{(i)}$. We then retain the merged TIV tokens $\bar{Z}_c$ and reorganize the representation of the entire video as:

$$\mathcal{Z} = \left[\bar{Z}_c, \ Z_{TV}^{(1,1)}, \ldots, Z_{TV}^{(1,T)}, \ \ldots, \ Z_{TV}^{(K,1)}, \ldots, Z_{TV}^{(K,T)}\right],$$

where $Z_{TV}^{(i,t)}$ denotes the TV tokens of the $t$-th frame in chunk $i$.

During decoding, following the broadcasting mechanism in Sec. 3.3, the shared tokens $\bar{Z}_c$ are broadcast to every frame to guide parallel reconstruction across all chunks. This design reduces token count by exploiting cross-chunk redundancy, improves efficiency by cutting complexity from quadratic to linear in $K$, and eases training by shortening token sequences.

## 4 EXPERIMENTS

### 4.1 IMPLEMENTATION DETAILS

Our model $\phi$ is optimized using a composite loss function that combines reconstruction quality with perceptual and adversarial objectives:

$$L = L_{\text{recon}} + \lambda_1 L_{\text{percept}} + \lambda_2 \cdot \lambda_\nabla L_{\text{adv}}, \quad \lambda_\nabla = \frac{\nabla_\phi (L_{\text{recon}} + \lambda_1 L_{\text{percept}})}{\nabla_\phi L_{\text{adv}}}, \tag{5}$$

This objective incorporates L1 reconstruction loss $L_{\text{recon}}$, perceptual loss $L_{\text{percept}}$ (Johnson et al., 2016; Larsen et al., 2016), and adversarial loss $L_{\text{adv}}$ (Goodfellow et al., 2020), with $\lambda_\nabla$ serving as an adaptive weighting coefficient. We empirically set $\lambda_1 = 1$ and $\lambda_2 = 0.2$ throughout our experiments. More implementation details can be found in Appendix B.

Table 1: **Comparison of Video Reconstruction on UCF-101.** We compare different categories of video tokenizers with similar compression ratios. We additionally report the number of tokens to pixels ratio (T/P (%)) for intuitive comparison, which is crucial for generation model efficiency. Gray highlights indicate cases where our method achieves superior or comparable performance. **Bold** values indicate best performance; underlined values show second-best results.

| Method | #Tokens | #Dim. | T/P(%)↓ | PSNR↑ | SSIM↑ | LPIPS↓ | rFVD↓ |
|---|---|---|---|---|---|---|---|
| *Downsample-based video tokenizer* | | | | | | | |
| SDXL-VAE (Podell et al., 2023) | 16384 | 4 | 1.563 | - | - | - | 23.68 |
| OpenSora (Zheng et al., 2024b) | 4096 | 16 | 0.391 | - | - | - | 67.52 |
| Cosmos-M (Agarwal et al., 2025) | 2048 | 16 | 0.195 | **31.70** | 0.9177 | 0.0575 | 13.67 |
| Cosmos-S (Agarwal et al., 2025) | 512 | 16 | 0.049 | 28.26 | 0.8577 | 0.1046 | 104.51 |
| CV-VAE (Zhao et al., 2024) | 4096 | 4 | 0.391 | 29.47 | 0.8849 | 0.0685 | 52.43 |
| *Holistic video tokenizer*(*:Video resolution 16×128×128) | | | | | | | |
| LARP (Wang et al., 2024a)* | 1024 | 16 | 0.391 | 28.65 | 0.9003 | **0.0425** | 23.93 |
| LARP (Wang et al., 2024a) | 1024 | 16 | 0.098 | 25.53 | 0.8262 | 0.0973 | 51.45 |
| ElasticTok (Yan et al., 2024) | 1024 | 16 | 0.391 | - | - | - | 390 |
| AdapTok (Li et al., 2025) | 2048 | 16 | 0.781 | 26.38 | 0.8539 | 0.0599 | 27.97 |
| *Decompose-based video tokenizer* | | | | | | | |
| Omni (Wang et al., 2024b) | 4096 | 8 | 0.391 | 29.34 | **0.9250** | 0.0487 | 14.53 |
| Omni-DV (Wang et al., 2024b) | 4096 | 8 | 0.391 | 28.06 | 0.9095 | 0.0637 | 27.12 |
| VidTwin (Wang et al., 2025) | 1008 | 4/8 | 0.126 | 28.14 | 0.8044 | 0.2414 | 388.86 |
| TivTok-S | 128 | 128 | **0.012** | 30.13 | 0.9010 | 0.0614 | 21.29 |
| TivTok-M | 512 | 32 | 0.049 | 30.26 | 0.8982 | 0.0533 | **12.65** |
| TivTok-L | 1024 | 16 | 0.098 | 29.54 | 0.8897 | 0.0607 | 17.97 |

## 4.2 VIDEO RECONSTRUCTION COMPARISON

We conduct comprehensive evaluation of video reconstruction quality on the UCF-101 (Soomro et al., 2012) dataset, utilizing videos with 256×256 resolution and 16-frame sequences. To ensure comprehensive comparison, we evaluate against representative baselines from three major categories: downsample-based, holistic, and decompose-based video tokenizers, with all methods configured to achieve similar compression ratios for meaningful comparison.

The quantitative results presented in Table 1 demonstrate that our method consistently achieves performance that either exceeds or matches current state-of-the-art approaches across all evaluation metrics. We further analyze the trade-off between the number of tokens and token dimensions by comparing TivTok-S, TivTok-M, and TivTok-L, which share the same overall model size. Interestingly, we observe that TivTok-M achieves the best reconstruction performance, suggesting that there exists an optimal balance between token number and dimensionality: too few tokens limit spatial resolution, while too low-dimensional tokens may restrict representational capacity. Nevertheless, the differences among the three models are relatively small, indicating that the framework is robust to this trade-off. Additionally, Table 5 compares tokenizers trained on different datasets (e.g., Web-Vid Bain et al. (2021)) while following the same content decomposition approach Yu et al. (2024b); Liu et al. (2025). The results show that our method achieves stronger compression, as it adopts a more general decomposition strategy rather than restricting factors to specific components (e.g., "motion" or "high-frequency"), which may not generalize to more complex video structures.

## 4.3 LONG VIDEO TOKENIZATION

To further demonstrate TivTok's superiority in temporal invariant reuse, we explore long video tokenization. The experimental results in Table 2 reveal distinct behavioral patterns as temporal length T increases. Downsample-based video tokenizers CV-VAE (Zhao et al., 2024) maintain relatively stable reconstruction quality but suffer from dramatic token count growth. Holistic video tokenizers LARP (Wang et al., 2024a) experience severe quality degradation while incurring quadratic latency scaling with respect to T. In contrast, our method achieves 2.91× higher compression efficiency while maintaining only slight reconstruction quality degradation and mitigating quadratic latency growth. Remarkably, our approach requires merely 1.1% of the tokens needed by downsample-based methods, demonstrating substantial potential for improving generation efficiency.

Table 2: **Comparison of long video tokenization.** We retrain baseline methods with comparable compression ratios and compare against CoordTok (Jang et al., 2025). We report results including inference latency for computational efficiency assessment.

| Method | #Tokens | #Dim. | Latency(s)↓ | PSNR↑ | SSIM↑ | LPIPS↓ | rFVD↓ |
|---|---|---|---|---|---|---|---|
| *Video resolution 32×256×256* | | | | | | | |
| CV-VAE (Zhao et al., 2024) | 8192 | 4 | 1.78 | 29.12 | 0.8809 | 0.0692 | 64.21 |
| LARP (Wang et al., 2024a) | 2048 | 16 | 1.75 | 23.15 | 0.7479 | 0.1757 | 226.79 |
| TivTok-S | 160 | 128 | **0.20** | 29.05 | 0.8831 | 0.0719 | 38.49 |
| TivTok-M | 640 | 32 | - | **30.25** | **0.8948** | **0.0591** | **23.26** |
| TivTok-L | 1280 | 16 | - | 29.13 | 0.8857 | 0.0711 | 61.46 |
| *Video resolution 128×256×256 (*:Video resolution 128×128×128)* | | | | | | | |
| CV-VAE (Zhao et al., 2024) | 32768 | 4 | 7.12 | **29.00** | **0.8831** | **0.0729** | **72.91** |
| LARP (Wang et al., 2024a) | 8192 | 16 | 22.78 | 14.85 | 0.2924 | 0.6251 | 3223.55 |
| CoordTok (Jang et al., 2025)* | 1280 | 8 | - | **27.25** | 0.7503 | 0.2346 | 1108.76 |
| TivTok-S | 352 | 128 | **0.71** | 26.23 | 0.8210 | 0.1057 | 92.09 |

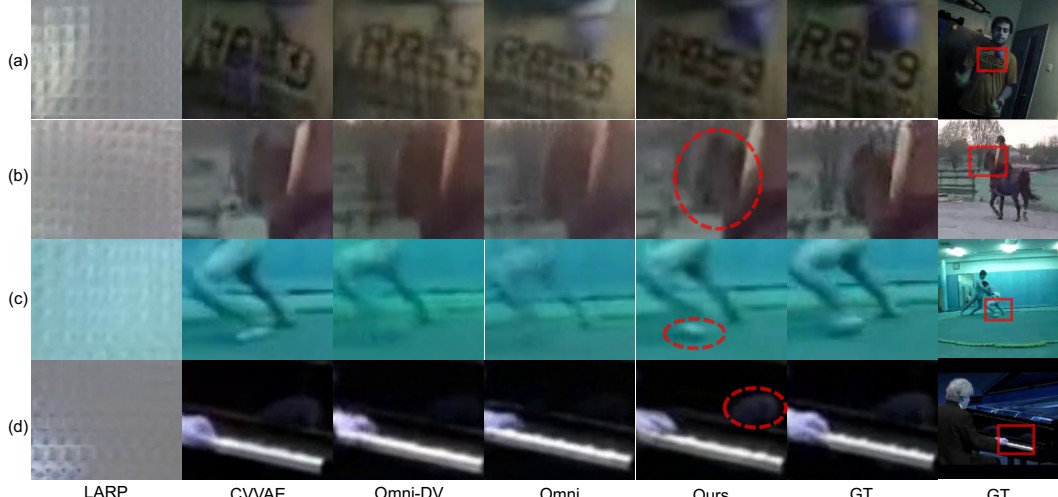

Figure 3: **Long Video Reconstruction Comparison on UCF-101.** We provide magnified views of regions highlighted by red rectangles in the ground truth for detailed comparison. Our method demonstrates superior detail preservation with higher compression, as indicated by the red circles.

We further provide qualitative visualization comparisons at a resolution of $128 \times 256 \times 256$ in Figure 3. Despite using a higher compression rate with fewer tokens, our method achieves comparable reconstruction quality and even superior detail preservation, as shown in Figure 3: the retained numerical text and ball in (a), the horse head in (b), the fine details around the foot in (c), and the subtle hand reflection on the piano surface in (d). These results demonstrate that TivTok's explicit extraction and reuse of temporal invariants substantially improve compression capability while maintaining strong reconstruction performance.

## 4.4 VIDEO GENERATION COMPARISON

Table 3 reports the generation metrics for class-conditional video generation on the UCF-101 dataset. The results show that our method achieves substantially lower GPU memory consumption and faster inference speed compared to the baselines, thanks to the reuse of TIV tokens, which greatly improves computational efficiency. At the same time, our generated videos remain highly competitive in quality under the same video length setting.

## 4.5 DISCUSSION ON TIME INVARIANT TOKENS

A key observation from our visualizations is that TIV tokens do not simply capture static backgrounds, but the true time-invariant components of a video. For example, in Figure 4(a) (the figure skating sequence), what changes across frames is primarily the background (advertising boards at

Table 3: **Comprehensive Comparison of Video Generation Methods.** The comparison includes inference speed, GPU memory usage, computational cost (TFLOPs), and generation quality (FVD). Results of MeBT Yoo et al. (2023), PVDM Yu et al. (2023), HVDM Kim et al. (2024), Coord-Tok Jang et al. (2025)+SiT-L/2 Ma et al. (2024) are taken from MALT Yu et al. (2025). (*: Video resolution 128×128×128).

| Method | Vid. Len. | #Tokens | Time / Step (s)↓ | GPU Peak Mem. (GB)↓ | TFLOPs↓ | FVD↓ |
|---|---|---|---|---|---|---|
| Cosmos-S | 16 | 512 | 0.047 | 2.62 | 0.49 | 191 |
| Omni | 16 | 4096 | 0.437 | 4.69 | 5.82 | 191 |
| LARP | 16 | 1024 | 0.083 | 2.73 | 1.05 | 107 |
| CVVAE | 16 | 4096 | 0.437 | 4.69 | 5.82 | 262 |
| TivTok-L | 16 | 1024 | 0.083 | 2.73 | 1.05 | **99** |
| TivTok-M | 16 | 512 | 0.047 | 2.62 | 0.49 | 101 |
| TivTok-S | 16 | **128** | **0.021** | **2.58** | **0.12** | 149 |
| CVVAE | 32 | 8192 | 1.261 | 10.82 | 15.97 | 370 |
| TivTok-S | 32 | **160** | **0.021** | **2.58** | **0.15** | **300** |
| MeBT* | 128 | 8192 | 6.53 | 13.3 | - | 968 |
| PVDM* | 128 | 16384 | 0.26 | 4.33 | - | 505 |
| HVDM* | 128 | 32768 | 1.514 | 12.1 | - | 550 |
| CoordTok+SiT-L/2* | 128 | 1280 | - | - | - | 369 |
| MALT* | 128 | 4096 | - | - | - | 220 |
| TivTok-S* | 128 | **352** | **0.031** | **2.60** | **0.33** | **208** |
| TivTok-S | 128 | **352** | **0.031** | **2.60** | **0.33** | 316 |

Table 4: **Ablation studies** on the proposed techniques.

| Methods | PSNR↑ | SSIM↑ | LPIPS ↓ | rFVD↓ |
|---|---|---|---|---|
| w/o decomposed tokens | 27.24 | 0.8530 | 0.0748 | 91.99 |
| w/o dual-scope encoder | 19.67 | 0.5691 | 0.5691 | 1359.38 |
| w/o parallel decoder | 17.69 | 0.4665 | 0.6083 | 3694.34 |
| w/o TIV-Broadcast training | 25.81 | 0.8219 | 0.1069 | 93.49 |
| TivTok | **29.05** | **0.8831** | **0.0719** | **38.49** |

different positions in the rink), while the two skaters remain consistent. The TIV tokens clearly capture the skaters' detailed appearance, especially the textural pattern of the clothing, indicating that they encode the essential invariant content rather than just static scenery. Interestingly, in some cases the time-invariant component is explicitly singled out by the model. In Figure 1 (bottom), although many balls move on the pool table, only the stationary ones are captured by TIV tokens; similarly, in the frisbee scene, the relatively immobile player is extracted as invariant. These cases show that TivTok selectively emphasizes elements that remain stable across time, rather than indiscriminately encoding all content.

More importantly, TIV tokens capture semantic invariants rather than only pixel-level persistences. To illustrate this, we compare against a simple intersection of unchanged pixels across frames (shown with red boxes), which would highlight only regions with minimal pixel variation. If TIV to-

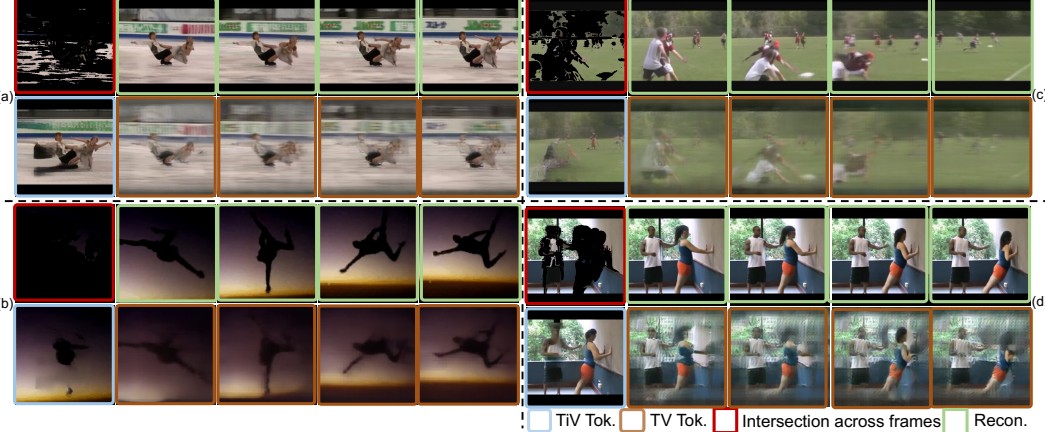

Figure 4: **TIV Token and TV Token Visualization and Analysis.** The intersection images (red boxes) display pixel-level persistence across frames, where we retain regions with minimal pixel variation. Results demonstrate that our TIV tokens capture temporal invariants including semantic information and scene geometry rather than merely pixel-level persistence.

Table 5: **Scalability of TivTok.** TivTok consistently improves with larger models and datasets, and maintains strong performance across different video resolutions.

| Method | Comp. Rate(%)↓ | PSNR↑ | SSIM↑ | LPIPS↓ | rFVD↓ |
|---|---|---|---|---|---|
| *Scalability of Model Size. Tested on UCF-101* | | | | | |
| TivTok-Small | 0.52 | 27.73 | 0.8611 | 0.0809 | 47.31 |
| TivTok-Base | 0.52 | 30.13 | 0.9010 | 0.0614 | 21.29 |
| TivTok-Large | 0.52 | 30.94 | 0.9116 | 0.0490 | 13.11 |
| *Scalability of Dataset Size and Resolution* | | | | | |
| CMD-WebVid-256 Yu et al. (2024b) | 6.85 | 26.55 | 0.795 | 0.110 | 98.623 |
| HiVAE-WebVid-256 Liu et al. (2025) | 0.27 | 29.35 | 0.834 | 0.096 | 61.941 |
| TivTok-WebVid-256 | 0.26 | 28.61 | 0.8288 | 0.0729 | 22.96 |
| TivTok-WebVid-256 | 0.52 | 31.69 | 0.8958 | 0.0477 | 7.15 |
| TivTok-VidProM-256 | 0.52 | 33.17 | 0.9384 | 0.0284 | 5.63 |
| TivTok-VidProM-512 | 0.52 | 33.56 | 0.9 | 0.0430 | 9.08 |

kens only encoded pixel-level persistence, they would align closely to these intersections. However, our results show otherwise: in cases such as the boxing sequences in Figure 1 and the dynamic scenes in Figure 4(b)/(c), TIV tokens encode stable scene semantics. For example, in the boxing sequence, beyond the moving punching bag, the entire gym environment—including the central pillar—is faithfully represented. This ability to capture semantic invariants provides strong reconstruction priors, leaving only minimal frame-specific details to be represented by TV tokens. Broadcasting TIV tokens supplies stable context across longer sequences, whereas broadcasting only pixel-level invariants would fail. The successful decoupling of TIV and TV tokens thus enables substantial redundancy reduction by exploiting the reusability of TIV tokens in longer videos.

### 4.6 ABLATION STUDY

Table 4 presents ablation studies on the $32 \times 256 \times 256$ setting. Removing token decomposition causes significant performance degradation, confirming that direct holistic tokenization complicates learning. Ablating TIV-Broadcast training reduces performance while maintaining capability, validating temporal invariant sharing. Ablating the specialized encoder or decoder causes complete failure, indicating that careful architecture design is required for effective broadcasting. All components are essential for effective long video compression.

### 4.7 SCALABILITY OF TIVTOK

Table 5 evaluates the scalability of our method with respect to model size, dataset size, and resolution. Performance consistently improves with larger models, reflecting their greater capacity to capture complex temporal and spatial patterns. Scaling up the dataset, e.g., using WebVid-10M Bain et al. (2021) and VidProM Wang & Yang (2024), further enhances the model's capabilities by providing more diverse training data. Our method also maintains strong reconstruction quality across different resolutions, demonstrating robustness. Overall, these results confirm that our approach scales well across model, dataset, and resolution dimensions, thanks to a streamlined design that includes only the essential modules for realizing our core ideas.

## 5 CONCLUSION

In this paper, we propose TivTok (*Time-Invariant Tokenizer*), which decouples videos into **time-invariant (TIV) tokens** capturing shared information and **time-variant (TV) tokens** encoding frame-specific details. TivTok employs a dual-range attention encoder and parallel decoder with TIV Token Broadcasting to isolate shared versus frame-specific content and enable token reuse across video. Experiments reveal that TIV tokens capture semantic information and scene geometry beyond pixel-level persistence, enabling natural extension to long videos. TivTok achieves superior reconstruction quality while delivering 2.91× compression efficiency improvement compared to state-of-the-art methods. By explicitly modeling temporal invariants and enabling their systematic reuse, TivTok establishes a new paradigm for efficient video tokenization that addresses the fundamental challenges of redundancy and scalability in video compression.

# 6 ETHICS STATEMENT

This work adheres to the ICLR Code of Ethics. No human subjects or animal experimentation was involved. All datasets used, including UCF-101 and K600, were sourced in compliance with relevant usage guidelines, ensuring no violation of privacy. We have taken care to avoid any biases or discriminatory outcomes in our research process. No personally identifiable information was used, and no experiments were conducted that could raise privacy or security concerns. We are committed to maintaining transparency and integrity throughout the research process.

# 7 REPRODUCIBILITY STATEMENT

We have made every effort to ensure that the results presented in this paper are reproducible. The experimental setup, including training procedures, model configurations, and hardware details, is described in detail throughout the paper. We have provided comprehensive implementation details of our TivTok framework to assist others in reproducing our experiments.

All datasets used in the paper, such as UCF-101 and K600, are publicly available, ensuring consistent and reproducible evaluation results. Training hyperparameters, loss function formulations, and architectural specifications are explicitly documented.

We believe these measures will enable other researchers to reproduce our work and further advance the field.

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

## A  ENTROPY ANALYSIS OF SHARED VS. INDEPENDENT FRAME ENCODING

Consider a video sequence $\{X_1, \ldots, X_T\}$. If each frame is encoded independently, the sequence entropy is

$$H_{\text{indep}} = \sum_{t=1}^{T} H(X_t) = \sum_{t=1}^{T} \big( I(X_t; C) + H(X_t \mid C) \big), \tag{6}$$

where $C$ denotes a shared component, $I(X_t; C)$ the mutual information between $X_t$ and $C$, and $H(X_t \mid C)$ the frame-specific residual information.

By explicitly modeling $C$, the sequence entropy decomposes as

$$H_{\text{shared}} = H(X_{1:T}) = H(X_{1:T} \mid C) + I(C; X_{1:T}) = \sum_{t=1}^{T} H(X_t \mid C, X_{1:t-1}) + I(C; X_{1:T}). \tag{7}$$

Since $H(X_t \mid C, X_{1:t-1}) \leq H(X_t \mid C)$, when strong temporal invariants exist, we have

$$H_{\text{indep}} - H_{\text{shared}} \geq \sum_{t=1}^{T} I(X_t; C) - I(C; X_{1:T}) \approx (T-1)I(C; X_{1:T}) \gg 0, \tag{8}$$

where the approximation assumes each frame contributes roughly equally to the shared information $C$.

**Conclusion:** explicitly capturing temporal redundancy allows for dramatically fewer tokens while maintaining representation quality.

## B  MORE IMPLEMENTED DETAILS

### B.1  TRAINING DETAILS

For the tokenizer implementation, our method is built upon SoftVQ-VAE (Chen et al., 2025a). All tokenizers are trained on a combination of UCF-101 (Soomro et al., 2012) and K600 (Carreira et al., 2018) datasets using a single node equipped with 8 A800 GPUs for 100,000 iterations, requiring approximately 1 day. For the second stage training targeting long video compression, we conduct additional training for 50,000 iterations. For video generation evaluation, we adapt LightningDiT-XL/1 (Yao et al., 2025) to support video generation, training for 100,000 iterations on 8 A800 GPUs over approximately 1 day.

Our tokenizer is built upon a ViT-based architecture (except for the scalability experiments, we use the ViT-Base model), while the generation model is based on LightningDiT. For generation, we evaluate class-conditional generation on UCF101. Tables 6 and 7 provide the detailed configurations of TivTok and LightningDiT, respectively.

Table 6: **Training configuration of TivTok.**

| Configuration | Value |
|---|---|
| video resolution | $256\times256$ |
| enc/dec hidden dimension | 768 |
| enc/dec #layers | 12 |
| enc/dec patch size | $4\times8\times8$ |
| enc/dec positional embedding | 3D RoPE (video) |
| optimizer | AdamW |
| weight decay | 1e-4 |
| optimizer momentum | $\beta_1, \beta_2 = 0.9, 0.95$ |
| global batch size | 64 |
| training steps | 100K for 16 frames and 50K for token brocasting training |
| base learning rate | 1e-4 |
| warmup steps | 5K |
| learning rate schedule | cosine |
| augmentation | horizontal flip, center crop |
| perceptual weight $\lambda_1$ | 1 |
| discriminator | DINOv2-S |
| discriminator weight $\lambda_2$ | 0.2 |
| discriminator start | 30K |
| discriminator LeCAM | 0.001 |

**Learnable token details.** Following existing image 1D tokenizers, we design the encoder to learn $N$ independent latent tokens, while the decoder relies on a single mask token. For the time-invariant and time-variant tokens, we distinguish them in the encoder using an attention mask. In the decoder, we adopt a frame-wise decoding strategy, where each frame is reconstructed using the time-invariant token together with the time-variant token corresponding to the current latent frame.

Table 7: **Training and inference configuration of LightningDiT-XL.**

| Configuration | Value |
|---|---|
| hidden dimension | 1152 |
| #heads | 16 |
| #layer | 28 |
| patch size | 1 |
| positional embedding | APE |
| optimizer | AdamW |
| weight decay | 0 |
| optimizer momentum | $\beta_1, \beta_2 = 0.9, 0.95$ |
| global batch size | 512 |
| training steps | 100K |
| base learning rate | 1e-4 |
| learning rate schedule | constant |
| augmentation | center crop |
| diffusion sampler | Euler |
| diffusion steps | 50 |
| CFG interval start | 0.1 |
| timestamp shift | 2 |

**Effect of different loss functions.** For completeness, we summarize the effects of commonly used loss functions in video tokenization:

- **L1 reconstruction loss** encourages accurate pixel-level reconstruction and is the most stable term for training tokenizers.

- **Perceptual loss** improves texture sharpness and semantic alignment by comparing features in a pretrained network, thereby mitigating over-smoothed outputs.

- **Adversarial loss** enhances realism and high-frequency details, though it typically contributes less to the overall bitrate–quality trade-off than the reconstruction loss.

### B.2 METRICS

For video reconstruction, our assessment employs established metrics including PSNR, SSIM (Wang et al., 2004), LPIPS (Zhang et al., 2018), and reconstruction FVD (rFVD) (Unterthiner et al., 2018). For video generation, we use generation FVD (gFVD) to assess the quality of generated video sequences.

### B.3 MORE TIV TOKEN ANALYSIS

We provide additional visualizations of TIV and TV tokens in Figure 5, complementing the analysis presented in Figure 4 for a more in-depth understanding of their behavior.

## C ANALYSIS OF MULTIPLE TIME-INVARIANT (TIV) TOKENS

Figure 8 and Table 8 present a detailed analysis of using multiple TIV tokens for video tokenization. As expected, increasing the number of TIV tokens generally improves reconstruction quality, since more tokens are available to encode information. However, this comes at the cost of reduced compression efficiency.

Specifically, using 4 TIV tokens leads to higher reconstruction quality than the baseline while moderately improving efficiency. In contrast, using a single TIV token maximizes compression efficiency (2.91×) while maintaining FVD at a level comparable to models with more TIV tokens. These results highlight the trade-off between reconstruction quality and compression rate and demonstrate

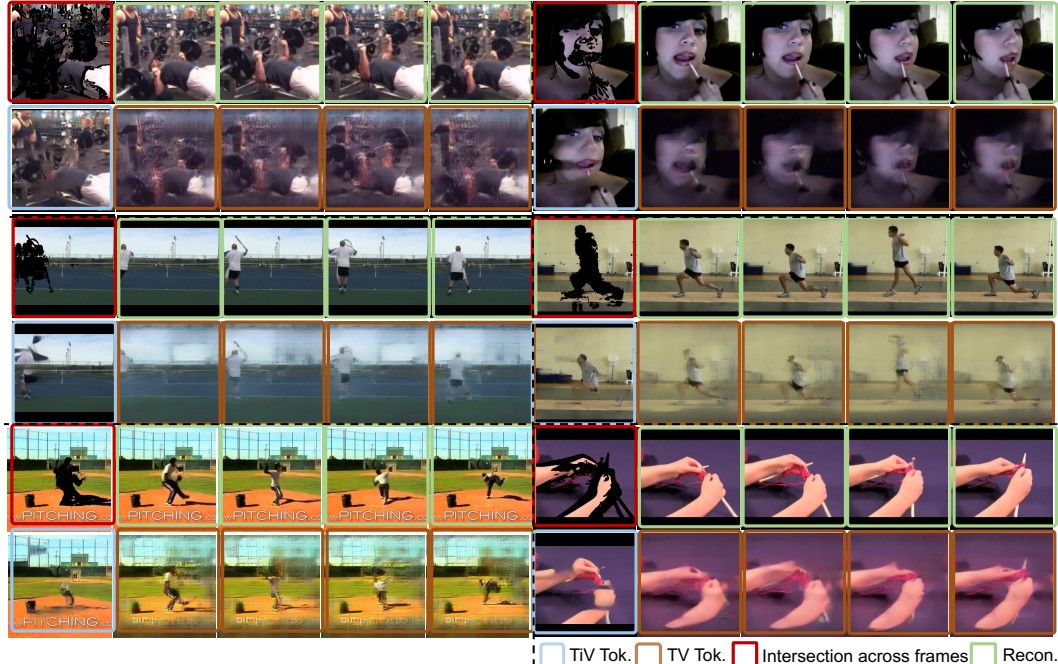

TiV Tok. | TV Tok. | Intersection across frames | Recon.

Figure 5: **More TIV Token and TV Token Visualization.** The intersection images (red boxes) display pixel-level persistence across frames, where we retain regions with minimal pixel variation. Results demonstrate that our TIV tokens capture temporal invariants including semantic information and scene geometry rather than merely pixel-level persistence.

Table 8: **Effect of the number of TIV tokens on reconstruction and compression metrics.** Experiments are conducted at 128×256×256 resolution.

| Method | Num TIV | Tokens | Dim | Comp. Ratio (%) | PSNR ↑ | SSIM ↑ | LPIPS ↓ | rFVD ↓ |
|--------|---------|--------|-----|-----------------|--------|--------|---------|--------|
| CVVAE  | -       | 32768  | 4   | 0.521           | 29.00  | 0.8831 | 0.0729  | 72.91  |
| TivTok | 8       | 1024   | 128 | 0.521           | 30.07  | 0.9003 | 0.0618  | 28.96  |
| TivTok | 4       | 640    | 128 | 0.326           | 28.97  | 0.8825 | 0.0739  | 39.84  |
| TivTok | 2       | 448    | 128 | 0.228           | 27.20  | 0.8453 | 0.0951  | 81.18  |
| TivTok | 1       | 352    | 128 | 0.179           | 26.23  | 0.8210 | 0.1057  | 92.09  |

the flexibility of our approach in adjusting the number of TIV tokens according to different application requirements.

Overall, this analysis confirms that multiple TIV tokens can be used to capture more detailed time-invariant content, but even a single TIV token effectively encodes the essential semantic invariants while providing strong compression.

## D  ANALYSIS OF TIV/TV TOKEN RATIO

Figure 7 and Table 9 present an analysis of the effect of different TIV-to-TV token ratios on reconstruction and compression performance. As expected, a smaller TIV/TV ratio tends to improve compression performance for long videos, because more TV tokens are used, which increases the total number of tokens. However, this also reduces efficiency.

Specifically, as the TIV/TV ratio decreases, reconstruction metrics such as PSNR and SSIM slightly decrease, while the compression ratio improves. Experimental results show that, regardless of the TIV/TV ratio, the compression efficiency of our method consistently surpasses CVVAE. Notably, when the TIV/TV ratio is set to 1:3 or 1:1, both reconstruction quality and compression performance

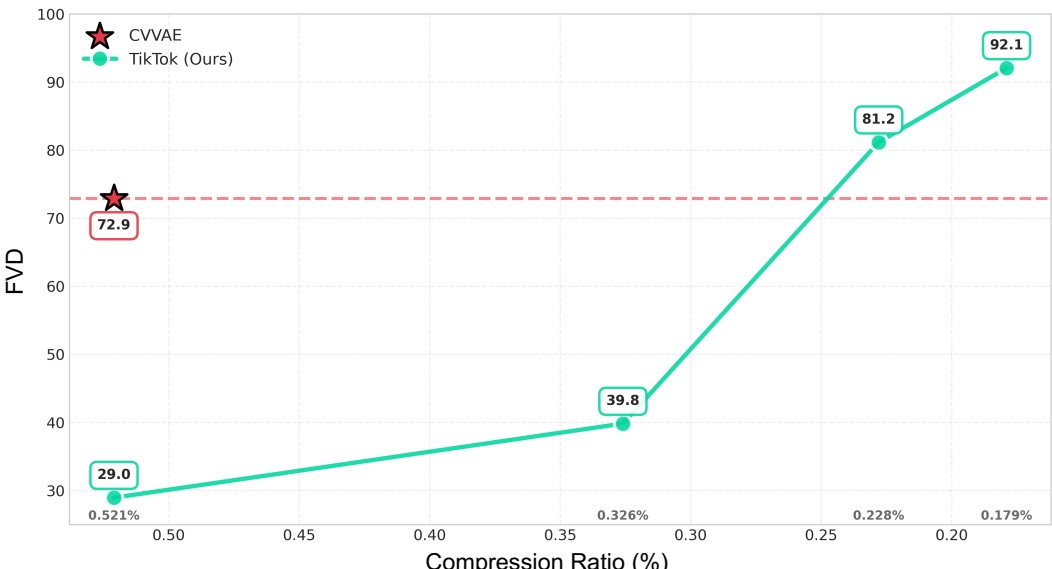

Figure 6: **Effect of multiple TIV tokens on reconstruction and compression.** This figure shows how varying the number of time-invariant (TIV) tokens impacts reconstruction quality (PSNR, SSIM) and compression ratio. More TIV tokens improve reconstruction but reduce compression efficiency, highlighting the trade-off between quality and token usage.

Table 9: **Effect of TIV/TV token ratio on reconstruction and compression metrics.** Experiments are conducted at 128×256×256 resolution.

| Method | TIV/TV Ratio | Compression Ratio | PSNR ↑ | SSIM ↑ | LPIPS ↓ | FVD ↓ |
|--------|--------------|-------------------|--------|--------|---------|-------|
| CVVAE | - | 0.521 | 29.00 | 0.8831 | 0.0729 | 72.91 |
| TivTok | 1:3 | 0.318 | 28.24 | 0.8663 | 0.0761 | 41.33 |
| TivTok | 1:1 | 0.229 | 27.52 | 0.8503 | 0.0887 | 64.76 |
| TivTok | 3:1 | 0.179 | 26.23 | 0.8210 | 0.1057 | 92.09 |

significantly outperform CVVAE. These results demonstrate a clear trade-off and provide practical guidance for selecting the TIV/TV token ratio based on specific application requirements.

Overall, this analysis confirms that adjusting the TIV/TV ratio allows flexible control over reconstruction versus compression, enabling the tokenizer to adapt to different video characteristics.

# E DECOMPOSITION DEMONSTRATION

To validate the decomposition property of our method, we conduct an experiment where the time-invariant (TIV) tokens are fixed and only the time-variant (TV) tokens are varied. As shown in Figure 8, this allows the model to generate different video sequences while keeping the shared content consistent, even when the subject undergoes significant motion. This behavior arises from our more general decomposition design, which enables TIV tokens to autonomously capture the information they consider more time-invariant, making it reusable across frames.

# F MORE QULITATIVE RESULTS

Figure 9 provides additional qualitative comparisons. Notably, CV-VAE operates at a compression rate of 0.521%, while Omni and Omni-DV both use a compression rate of 1.04%. In contrast, our method achieves a substantially lower compression rate of only 0.179% by reusing TIV tokens. Despite this significantly more constrained setting, our method produces visual results that remain comparable to existing approaches.

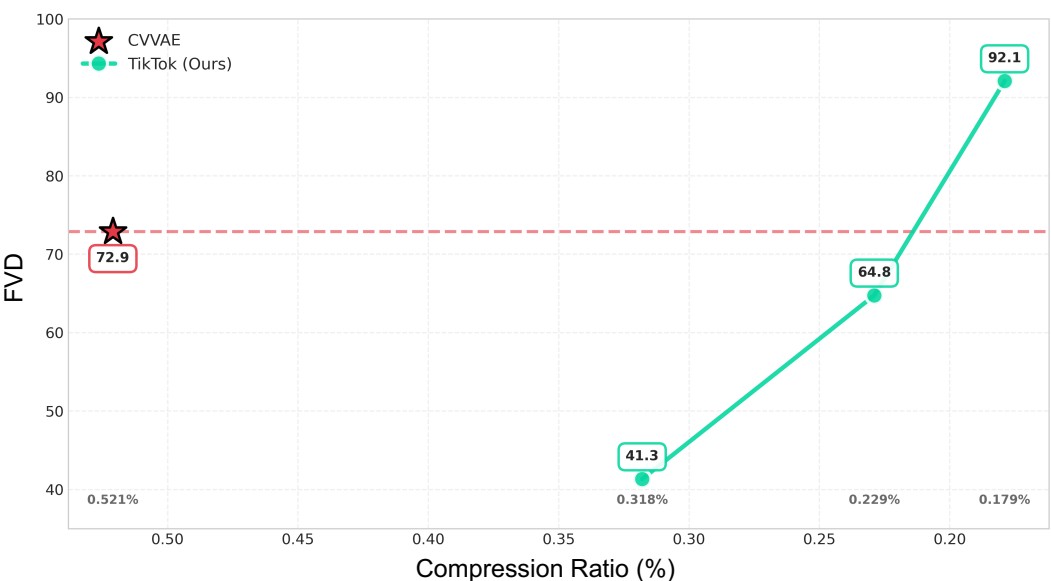

Figure 7: **Effect of different TIV-to-TV token ratios on reconstruction and compression performance.** The results illustrate the trade-off between reconstruction quality and compression efficiency.

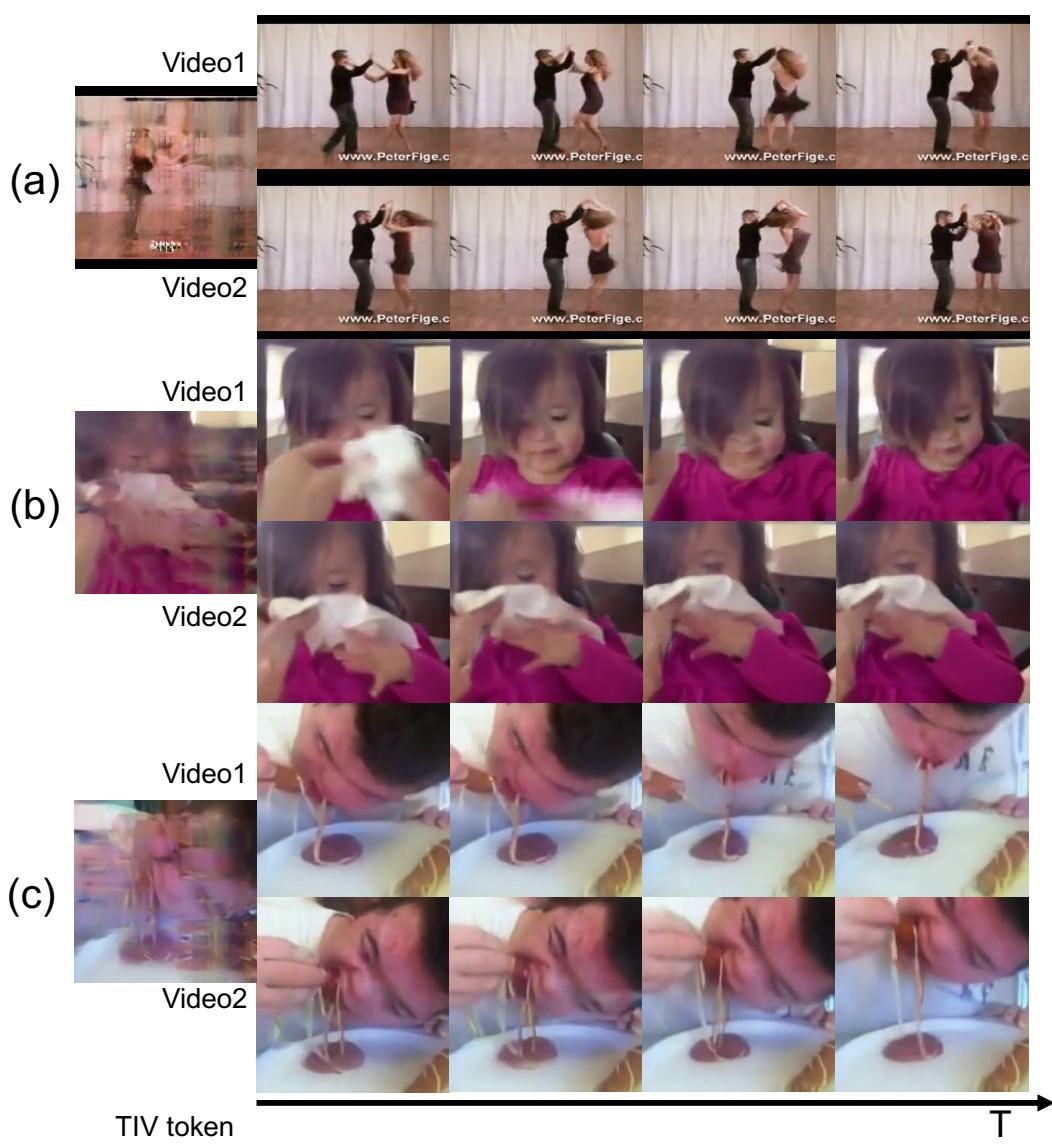

Figure 8: **Demonstration of the decomposition property.** Time-invariant (TIV) tokens are fixed while time-variant (TV) tokens are varied, showing that the model can generate different video sequences while preserving shared content.

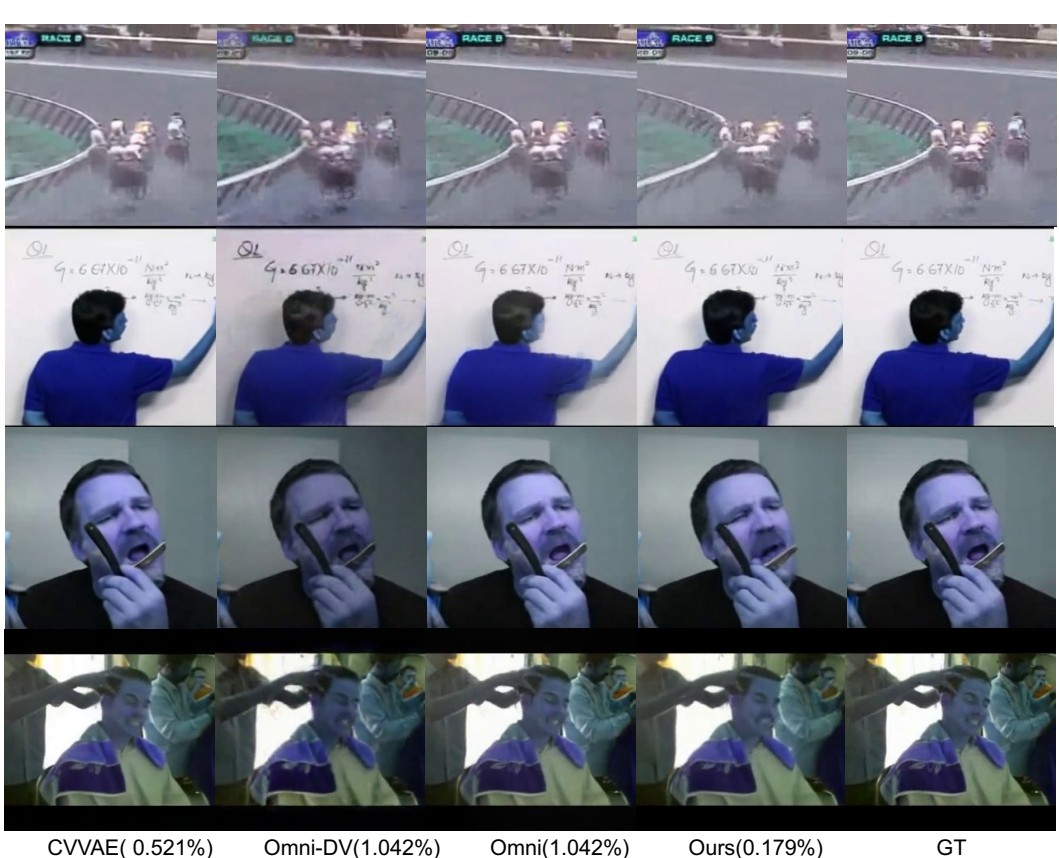

CVVAE( 0.521%)    Omni-DV(1.042%)    Omni(1.042%)    Ours(0.179%)    GT

Figure 9: **Additional qualitative examples.** Compression ratios for each method are shown in parentheses. Despite operating at a significantly lower compression ratio, our method produces visual results that remain comparable to existing approaches.

