# OpenReview forum: "TivTok: Broadcasting Time-Invariant Tokens for Scalable Video Tokenization"
_ICLR.cc/2026/Conference — Submitted to ICLR 2026_

### Official Review · Reviewer_VFj8 · 2025-10-28

**Soundness:** 3
**Presentation:** 3
**Contribution:** 3
**Rating:** 6
**Confidence:** 5

**Summary:**

TIVTok provides a way to take advantage of redundancy in video tokenization (for generative models). Typically prior works consider pixel-level redundancy; however, TIVTok instead proposes a notion of semantic redundancy and lets the model learn exactly what to persist across each frame. It does so by only allowing the TIV tokens to attend to every frame, while letting other tokens attend to others only within the same frame. This forces the model to use the TIV tokens for higher-level concepts that persist across the entire video, and the per-frame TV tokens to be used for lower-level details. The results show that this method is able to perform on par with other video tokenizers while compressing the input into significantly less tokens.

**Strengths:**

1. I think the idea is very clever. Redundancy has generally been limited to pixel-level (spatial / temporal redundancy) rather than semantic, and the TIV tokens are a nice way to incorporate this.
2. I think the mechanism of using the invariant tokens and keeping time-variant tokens separate per frame is a nice way to parallelize the decoding and potentially decrease the memory and time usage while also making generation potentially faster.
3. I think the paper is clearly written and it is very easy to grasp the novelty of the method and what the core contribution is.

**Weaknesses:**

1. The paper never measures the reduction in wall-clock encoding and decoding time, or the memory savings that come from using the time invariant tokens, and only reports TFLOPS instead. I think memory / WC time are much more important and useful metrics to report, so I would want to see those.
2. A key design decision in this paper is deciding how much time invariant tokens to use and understanding exactly what they capture. As far as I can tell, there is no systematic evaluation of the tradeoff between the number of TIV tokens and TV tokens, and the analysis that measures the effect of the TIV tokens is quite confusing. For example, it's not obvious at all from Figure 4 that the ice skater is covered by the TIV tokens - wouldn't you have to do something like masking out the TV tokens, or vary them in some way to understand the the effect of these? What does "intersection across frames" mean?
3. The whole entropy discussion in the methods section seems very unnecessary. It's obvious that if there is a lot of repeated information content that the entropy of the video will be less - this math is not providing any new insight and it would be better served to have some additional analysis from the experiments instead.
4. The generation experiment seems to be somewhat of an afterthought, and it's not clear to me why TIVTok would necessarily lead to a better representation - there are very few baselines here and the experiment is not explained in any detail.  I think the generation aspect should be explored more, and if it were very clear that TIVTok led to a generation quality that was no worse and also more efficient (due to less tokens + parallelizable per frame attention) then it would be a stronger contribution.

**Questions:**

1. My main suggestion is that i think seeing the decrease in memory and WC time would be very helpful to understand the impact of this method.
2. I would want a clearer explanation of the analysis on what the TIV tokens actually are doing, not necessarily a new experiment but just a clarification on why it's obvious that the TIV tokens actually learn semantic invariants. If so, I would gladly raise my rating.

---

> ### Author Response · Authors · 2025-11-21
>
> We sincerely thank reviewer VFj8 for the valuable time and for recognizing our idea, methodology, and writing. We also truly appreciate the thoughtful comments regarding the practical performance and insights of our approach, which will further help us improve the quality of our work. Below, we provide our responses to each of the concerns raised.
>
> > W1 & Q1: Wall-clock encoding and decoding time and memory savings with time invariant tokens.
>
> We thank the reviewer for highlighting this important point. In the revised manuscript, we have supplemented the evaluation with wall-clock encoding/decoding time and GPU peak memory, which are now reported in Table 3 and in the additional results provided below.
>
> | Method      | Vid. Len. | #Tokens | Time / Step (s) ↓ | GPU Peak Mem. (GB) ↓ | TFLOPs ↓ | FVD ↓   |
> |------------|------------|---------|------------------|---------------------|----------|---------|
> | Cosmos-S   | 16         | 512     | 0.047            | 2.62                | 0.49     | 191     |
> | Omni       | 16         | 4096    | 0.437            | 4.69                | 5.82     | 191     |
> | LARP       | 16         | 1024    | 0.083            | 2.73                | 1.05     | 107     |
> | CVVAE      | 16         | 4096    | 0.437            | 4.69                | 5.82     | 262     |
> | CVVAE      | 32         | 8192    | 1.261            | 10.82               | 15.97    | 370     |
> | TivTok-L   | 16         | 1024    | 0.083            | 2.73                | 1.05     | 99      |
> | TivTok-M   | 16         | 512     | 0.047            | 2.62                | 0.49     | 101  |
> | TivTok-S   | 16         | 128     | 0.021            | 2.58                | 0.12     | 149     |
> | TivTok-S   | 32         | 160     | 0.021            | 2.58                | 0.15     | 300     |
> | TivTok-S   | 128        | 352     | 0.031            | 2.60                | 0.33     | 316     |
>
> Thanks to the use of time-invariant tokens, our method requires significantly fewer tokens for long-video generation, leading to clear reductions in both inference time and GPU memory consumption compared to existing approaches.
>
> Furthermore, we extend the latency analysis of Table 2 by incorporating both encoding and decoding time as well as GPU memory usage, providing a more comprehensive and practically relevant comparison below. The results indicate that our method achieves higher efficiency and significantly lower GPU memory consumption than the baselines.
>
> | Method   | Frames | Encoding Time (s) | Decoding Time (s) | Total Time (s) | Encode GPU Peak Memory (GB) | Decode GPU Peak Memory (GB) |
> |----------|--------|------------------|------------------|----------------|----------------------------|----------------------------|
> | CVVAE    | 16     | 0.1029           | 0.1764           | 0.2793         | 2.3126                     | 2.5245                     |
> | CVVAE    | 32     | 0.1949           | 0.3522           | 0.5471         | 2.3303                     | 2.5426                     |
> | CVVAE    | 128    | 0.7383           | 1.4051           | 2.1434         | 2.4362                     | 2.6508                     |
> | TivTok   | 16     | 0.0263           | 0.0114           | 0.0376         | 0.4904                     | 0.4944                    |
> | TivTok   | 32     | 0.0453           | 0.0187           | 0.0640         | 0.5925                     | 0.6006                    |
> | TivTok   | 128    | 0.1533           | 0.0652           | 0.2185         | 1.1984                     | 1.2323                    |
>
> > W2-1: Systematic Evaluation of TIV-to-TV Token Ratio.
>
> We thank the reviewer for highlighting this important design consideration. We have supplemented this analysis in **Appendix D**. **Figure 7 and Table 9** provide a systematic evaluation of the trade-off between the number of time-invariant (TIV) tokens and time-variant (TV) tokens. As detailed in the analysis, adjusting the TIV/TV ratio allows flexible control over reconstruction quality versus compression performance. A smaller TIV/TV ratio tends to improve compression for long videos, at the cost of slight reductions in PSNR and SSIM, while a larger ratio favors reconstruction quality. **Notably, when the TIV/TV ratio is set to 1:3 or 1:1, both reconstruction quality and compression performance significantly outperform CVVAE.** These results clearly demonstrate the advantages of our method in extracting and reusing TIV tokens to achieve superior reconstruction and compression efficiency.

---

> ### Author Response · Authors · 2025-11-21
>
> > W2-2 & Q2: What TIV token exactly captures and more illustrations of Figure 4.
>
> We thank the reviewer for raising this important question and for the opportunity to clarify our analysis. In Figure 4, “intersection across frames” (red box) refers to the set of pixels that remain stable throughout the video. Concretely, we take the first frame as a reference and retain only the pixels whose intensity variation across selected key frames is below a small threshold (50 in our implementation). This produces a visualization of pixel-level persistence in the raw video. The green box shows the reconstruction of the full video using both TIV and TV tokens. To analyze the role of each type of token, we perform token-masked reconstructions: Blue box: reconstruction only using TIV tokens while masking out all TV tokens. Brown box: reconstruction only using TV tokens while masking out all TIV tokens. This controlled masking directly isolates the contribution of each token type and avoids the ambiguity the reviewer was concerned about.
>
> From these masked reconstructions, we observe: TIV tokens do more than capture pixels that remain constant across frames—they infer what is semantically time-invariant in the video. While they naturally include regions highlighted by the pixel-intersection map, their behavior extends well beyond simple low-variance pixels. For example, in the boxing scene in Figure 1 (top), the swinging punching bag intermittently occludes the pillar and window behind it. Although these regions are not pixel-constant—their visible appearance changes due to occlusion—the underlying semantics (the presence of the pillar and window) remain time-invariant. TIV tokens successfully recover these structures by integrating partial evidence across frames, effectively completing the full pillar and window. This demonstrates that TIV tokens capture semantic, scene-level invariants, rather than merely tracking pixel-level consistency.  Figure 4(d) shows a similar phenomenon, where semantically stable background elements are reconstructed despite pixel variations. In Figure 4(a), although the ice skater’s limb positions vary over time, the textural pattern of the clothing remains invariant, and the TIV tokens accurately preserve these details as well.
>
> What TIV tokens are actually doing and why. The special behavior of TIV tokens arises from our dual-range attention mask in the encoder, which explicitly guides the flow of information, combined with TIV token broadcasting in the decoder, ensuring that each frame’s reconstruction relies on the TIV tokens. For tokenization that is an inherently under-determined problem, more information must be compressed into fewer tokens, so each token is incentivized to maximize the information it carries. Because TIV tokens are used across all frames, they naturally first encode content present in every frame. With their remaining capacity, the most efficient strategy is to capture information appearing in multiple frames—even if not all pixels are visible at once. For example, in the boxing scene, the pillar behind the punching bag may be partially occluded in some frames, but different parts appear in others. TIV tokens integrate these partial observations to encode the full structure, which is semantically invariant across time.
>
> In summary, TIV tokens’ global view allows them to prioritize highly redundant information, enabling them to capture time-invariant content that is reusable and providing a clear advantage for long-video tokenization and compression.
>
> > W3: Revision of the Entropy Discussion.
>
> We thank the reviewer for the constructive comment. The entropy discussion was included to provide a systematic justification for the motivation behind our decomposition strategy. While the conclusion—that independently encoding frames consumes significantly more tokens than separating shared content from frame-specific residuals—may indeed be intuitive to experienced readers, the derivation helps formalize this intuition and clarify the underlying assumptions. In the revised manuscript, we have streamlined this section in the main text and moved the full derivation to the Appendix, keeping the paper concise while still offering a complete explanation for readers who are interested in the theoretical details.

---

> ### Author Response · Authors · 2025-11-21
>
> > W4: More generation experiment details and results.
>
> We thank the reviewer for the insightful comments. In the revision, we have expanded the generation experiments in Table 3 to address these concerns. **Specifically, we include long-video generation results and additional comparisons in terms of inference speed and GPU memory usage.** These new experiments demonstrate that TivTok substantially reduces the number of required tokens during generation, which directly leads to much higher efficiency. Moreover, the FVD metrics show that our generation quality remains comparable to the baselines, despite the improved efficiency. Overall, these results support the claim that TivTok provides an effective and efficient representation for video generation. Notably, we also provide additional implementation details for generation in the **Appendix B**.

---

### Official Review · Reviewer_F2Sa · 2025-10-29

**Soundness:** 2
**Presentation:** 3
**Contribution:** 3
**Rating:** 6
**Confidence:** 5

**Summary:**

This paper presents a new tokenization method for videos, called TivTok. The main idea is to decompose video tokens into (1) time-invariant tokens that are shared across all video frames and (2) time-variant tokens that are allocated to each frame. Intuitively, the time-invariant tokens encode static features in videos (e.g., background), while the time-variant tokens capture motion information. To achieve this, TivTok extends TiTok, an efficient image tokenization method that uses two different types of tokens. The authors also modify the encoder’s attention mask: the time-invariant tokens attend to all tokens across all frames, whereas the time-variant tokens attend only to the current frame and the time-invariant tokens. For the decoder, they introduce parallel decoding, where each frame is decoded independently using the corresponding time-invariant and time-variant tokens. By doing so, the proposed method achieves better reconstruction quality than other video tokenizers under the same compression ratio.

**Strengths:**

- The paper is generally well-written and easy to follow.
- I like the idea of decomposing videos into time-invariant parts and time-variant parts. There have been similar approaches, but I think this approach tries to do a more explicit decomposition than prior works, as well as showing better reconstruction quality.
- The paper seems to have a great compression ratio compared with existing video compression methods.

**Weaknesses:**

Although I appreciate the idea introduced in this paper, I have several concerns:
- **Scalability of the tokenizer.** The paper only conducts experiments on UCF-101 with a resolution of 256×256. While I understand that training a video tokenizer requires significant computational resources, it is still crucial to demonstrate scalability in terms of dataset size or resolution. Could the authors provide additional results by training the tokenizer on larger and higher-resolution datasets, such as VidProM [1]? If memory is the bottleneck, one possible workaround is to first encode each frame into latent vectors using SD-VAE or other image VAEs, and then reconstruct it by training TiVTok in that latent space. The framework could follow the structure: SD-VAE Encoder → TiVTok → SD-VAE Decoder.
- **Evaluation beyond reconstruction.** Showing only reconstruction quality is insufficient, as this paper focuses on tokenization rather than compression. While the authors do provide generation results on UCF-101, many implementation details are missing. For example, is the generation class-conditional? What is the video length used for training the generative model? Moreover, can this tokenizer be applied to other downstream tasks, such as video question answering (VQA)?
- **Demonstrating decomposition.** To validate the claimed decomposition property, it would be valuable to include generation results where the time-invariant tokens are fixed and only the time-variant tokens are varied. This would help visualize the separation between static and dynamic information, similar to the experiment presented in Appendix F of the CMD paper [2].

[1] https://vidprom.github.io/
[2] Yu et al., Efficient Video Diffusion Models via Content-Frame Motion-Latent Decomposition, ICLR 2024

**Questions:**

- Could the authors provide more details about the video generation experiments?
- The proposed trick for extending to “long videos” might have a practical limit on the maximum number of frames it can handle. For example, it could be difficult to process 1000 frames, since in that case there may be no time-invariant component shared across all frames. Could the authors provide any insights or empirical results regarding this limitation?

---

> ### Author Response · Authors · 2025-11-21
>
> We sincerely thank reviewer F2Sa for the valuable time and for recognizing our idea and experimental results. We also truly appreciate the thoughtful comments regarding scalability and downstream tasks, which will further help us improve the quality of our work. Below, we provide our responses to each of the concerns raised.
>
> > W1: Scalability of the tokenizer.
>
> We sincerely thank the reviewer for this insightful suggestion, which has helped us improve the paper. In the revised manuscript, **Table 5** provides a comprehensive study of the scalability of our tokenizer, including model size, dataset size, and resolution.
>
> Regarding VidProM, this dataset provides both a larger number of videos and higher-resolution frames. We successfully validated the scalability of our method on VidProM. Some key results are summarized below, demonstrating that our approach maintains strong reconstruction quality and efficiency even at larger scales and higher resolutions.
>
> We believe these results strongly support the generalizability and practical applicability of our method to larger datasets and higher resolutions.
> | Setting                |  PSNR  |   SSIM   |  LPIPS  |  FVD  |
> |------------------------|--------|----------|---------|-------|
> | TivTok-VidProM-256     | 33.17  | 0.9384   | 0.0284  | 5.63  |
> | TivTok-VidProM-512     | 33.56  | 0.9374   | 0.0430  | 9.08  |
>
> > W2-1 & Q1: More Generation Details.
>
> We thank the reviewer for pointing out this important issue and helping us improve the paper. We have added detailed experimental settings for video generation in the **Appendix B**. The generation experiments are class-conditional, and in the previous version, the training video length was 16 frames.
>
> In the revised version, we further extend these experiments by including longer videos and additional efficiency comparisons. We strongly encourage the reviewer to refer to Table 3 in the updated manuscript for a comprehensive overview.
>
> > W2-2: More downstream applications.
>
> We thank the reviewer for the question regarding potential downstream applications such as video question answering (VQA). Our current tokenizer is primarily trained for reconstruction and generation tasks, focusing on capturing both time-invariant and time-variant information to enable efficient video reconstruction.
>
> VQA, in contrast, requires semantic understanding and reasoning over video content rather than pixel-level reconstruction. Therefore, a tokenizer trained solely with a reconstruction objective is unlikely to provide features sufficiently aligned with the needs of VQA.
>
> However, our approach could potentially be adapted for such tasks. For example, one could fine-tune the pretrained tokenizer on VQA datasets using task-specific objectives, or combine our tokens with a separate semantic embedding network that captures higher-level concepts, enabling the model to leverage both detailed reconstruction features and semantic understanding. We leave this as an interesting direction for future work.
>
> > W3: Decomposition Demonstration.
>
> We thank the reviewer for the valuable suggestion. Following this, we have added the corresponding experiment in **Figure 8 of Appendix E**. By fixing the time-invariant tokens and varying the time-variant tokens, our method is able to generate different videos, even when the subject motion is relatively large. This is due to our more general decomposition approach, which allows the TIV tokens to autonomously capture the information they consider more time-invariant, making this part of the representation more reusable across frames.
>
> > Q2: Extra long video tokenization.
>
> We thank the reviewer for raising this important question regarding encoding very long videos, which remains largely **under-explored** before. For extremely long videos, e.g., 1000 frames, the situation can vary depending on the content. On one hand, for videos with relatively little change, such as a fixed camera shot, even 1000 frames may still contain substantial time-invariant content, allowing our method to effectively compress the video. On the other hand, for videos with significant content changes, such as multiple scene cuts, the global time-invariant component may indeed be limited. In such cases, a **divide-and-conquer** approach can be adopted: multiple TIV tokens can be used to capture the time-invariant content for each scene separately, reducing redundancy in each segment.
>
> Overall, our method provides a new insight compared to prior approaches by decomposing videos into time-invariant and time-variant parts, rather than strictly pre-defining two components. This decomposition allows flexible reuse of TIV tokens, enabling adaptation to different types of videos and varying levels of temporal redundancy.

---

> > ### Comment · Reviewer_F2Sa · 2025-11-22
> > **Response**
> >
> > Thanks for the response. Most of my concerns are now addressed. With these experiments, I think the paper is strengthened more, and now I think this is a quite good paper. Regarding Table 3, could the authors separate FVD by length by adding some baselines? Because FVD is sensitive to the length, the current table might be misleading. You can consider adding some of the baseline results reported in Table 1 in [1].
> >
> > [1] MALT Diffusion: Memory-Augmented Latent Transformers for Any-Length Video Generation, CVPRW 2025

---

> ### Author Response · Authors · 2025-11-25
>
> We sincerely thank you for your thoughtful feedback and for engaging with our responses, and we greatly appreciate your positive recognition of our work. Following your advice, we have updated the manuscript and added additional baseline results to Table 3. Specifically, we incorporated the baselines reported in Table 1 of [1] to better disentangle the effect of video length on FVD and to avoid potential misinterpretation. For baselines whose original evaluation resolution differs from ours, we additionally evaluated our method under their resolution setting to ensure a fair comparison.
>
> We genuinely appreciate the reviewer’s careful reading and constructive feedback, which have substantially improved both the clarity and completeness of our paper.
>
> [1] MALT Diffusion: Memory-Augmented Latent Transformers for Any-Length Video Generation, CVPRW 2025

---

> > ### Comment · Reviewer_F2Sa · 2025-11-25
> > **Response**
> >
> > Thanks! I think the paper is much stronger after revision. Long video generation experiments and larger-scale tokenizer training really look great. I want to champion this paper and raise the score accordingly!

---

### Official Review · Reviewer_hY2j · 2025-10-31

**Soundness:** 3
**Presentation:** 2
**Contribution:** 2
**Rating:** 4
**Confidence:** 4

**Summary:**

In this paper, a transformer-based tokenizer called TivTok is proposed for more efficient video tokenization. Tivtok decomposes a video into time-invariant tokens and time-variant tokens. To this end, the encoder uses masked attention for time-variant tokens to enforce the factorization. In decoding, the time-invariant tokens are re-used for all frames to reduce the computation complexity. Also, the proposed algorithm is extended to the chunked compression scheme for the long video processing. Experimental results show that the proposed algorithm achieves comparable performances with the conventional methods with lower computation budget.

**Strengths:**

- This paper clearly describes the proposed algorithm and it is easy to follow.
- The motivation is solid and the proposed algorithm seems technically sound.
- The proposed algorithm shows decent video construction performance with lower computation budget.

**Weaknesses:**

- This paper does not provide sufficient details to reproduce the proposed method. For example, there is no detailed description of the encoder and decoder architecture. It would be helpful if such information were included, at least in the appendix.
- From Equations (5) and (6), it seems that the time-variant token may not encode only the frame-local residuals, as stated in L259. Ideally, that would be the case, but in practice, it might redundantly encode information that is already captured by the time-invariant tokens. While minimizing this redundancy would be more efficient and help reduce the loss, the claim might be somewhat overstated. Do the authors have any additional analysis related to this?
- It would be helpful to include a discussion on the performance differences among the S, M, and L models in Table 1.
- In Table 1, what is the difference between two LARP models in L337 and L338?
- Is there a reason why the performances of all S, M, and L models are not compared in Tables 2 and 3?
- How would the performance change if multiple time-invariant tokens were used — for example, by dividing the video into chunks without averaging the shared tokens?
- Typo
  - L149: complexities.In -> complexities. In
  - L185: , Producing -> , producing
  - L195 Eq1: Patchfy -> Patchify
  - L229: time-invarian -> time-invariant
  - L348: we report -> We report
  - L397: proposed techniques -> proposed techniques.

**Questions:**

Please find my concerns in the weakness section. I believe this paper has some strengths, and I am willing to increase my score if the concerns are properly addressed in the rebuttal.

---

> ### Author Response · Authors · 2025-11-21
>
> We sincerely thank reviewer hY2j for the valuable time and for recognizing both the motivation and the proposed algorithm. We truly appreciate the thoughtful comments and the detailed suggestions, which will further help us improve the quality of our work. Below, we provide our responses to each of the concerns raised.
>
> > W1: More details for reproducing the proposed method.
>
> We sincerely thank the reviewer for pointing out this shortcoming. Our tokenizer is fully implemented using a ViT-based architecture, and we have now supplemented the manuscript with additional details, including all relevant parameter configurations, in the **Appendix B**. If there are any remaining questions or clarifications needed, we would be happy to provide further information.
>
> > W2: Potential overstatement of TV-token disentanglement and more additional analysis.
>
> We sincerely thank the reviewer for the careful reading and for pointing out this important clarification that helps us further improve the rigor of our paper. We agree that the use of the word “only” in the original statement was overly strong, and we have revised the wording accordingly in the updated manuscript.
>
> Although we design the dual-range attention mask to separate the receptive fields of the TIV and TV tokens—encouraging TIV tokens to capture sequence-level shared information while guiding TV tokens toward frame-local residuals—the task itself is a pixel-level reconstruction task. As a result, both types of tokens ultimately optimize for faithful pixel reconstruction. In real video data, even for the same object, factors such as illumination fluctuations, shading, or complex motion often make it difficult for all invariant information to be captured exclusively by the TIV tokens. Consequently, some degree of redundancy is unavoidable, and certain information may still appear in the TV tokens. Achieving perfect disentanglement is inherently challenging in this setting.
>
> However, our design effectively encourages the intended separation. As shown in Figure 4,
>
> - in (a), the TIV token clearly captures stable elements such as the skater’s clothing, while the TV token focuses on background color changes and leaves the clothing mostly blurry;
>
> - in (d), the TIV token faithfully encodes wall structures and lines, whereas the TV token concentrates on the subject’s motion and largely ignores the wall.
>
> These visualizations support our claim that the dual-range masking successfully encourages TIV tokens to aggregate shared, time-invariant information across the sequence and guides TV tokens to focus primarily on frame-local residuals, even though full separation is theoretically difficult. We have revised the manuscript to clarify this nuance and avoid overstating the degree of disentanglement.
>
> > W3: Discussion on the performance differences among the S, M, and L models in Table 1.
>
> We thank the reviewer for this insightful question. First, we clarify that TivTok-S, TivTok-M, and TivTok-L have the same overall model size, and the only difference lies in the trade-off between the number of tokens and token dimensions.
>
> These models are designed to compare how token number vs. token dimension affects reconstruction and generation under the same absolute compression rate, rather than to study model scalability. From the experimental results:
>
> - Reconstruction: We observe that TivTok-M achieves the best reconstruction performance, suggesting that there exists an optimal balance between token number and dimensionality: too few tokens limit spatial resolution, while too low-dimensional tokens may restrict representational capacity. Nevertheless, the differences among the three models are relatively small, indicating that the framework is robust to this trade-off.
>
> - Generation: As shown in Table 3, a lower token dimension benefits DiT modeling, which aligns with empirical observations.
>
> Additionally, to provide a scalability comparison, we have included further scalability experiments. The results are presented below. The experiment shows that our method benefits from larger model configurations and achieves progressively better performance as the model size increases.
> | Model      |  PSNR  |   SSIM   |  LPIPS  |  FVD   |
> |------------|--------|----------|---------|--------|
> | Vit-Small  | 27.73  | 0.8611   | 0.0809  | 47.31  |
> | Vit-Base   | 30.13  | 0.901    | 0.0614  | 21.29  |
> | Vit-Large  | 30.94  | 0.9116   | 0.049   | 13.11  |
>
> >W4: Difference between two LARP models in L337 and L338 in Table 1.
>
> We apologize for the omission of details in the paper and thank the reviewer for pointing this out. The original LARP experiments were conducted at a resolution of 16×128×128, which is easier than the commonly used 16×256×256. Thus, L337 and L338 correspond to these two resolutions, respectively. We have clarified this in the revised version of the paper.

---

> ### Author Response · Authors · 2025-11-21
>
> > W5: Reason for Missing M and L Models in Tables 2 and 3.
>
> As discussed in our response to W3, the reconstruction performance of TivTok-S, TivTok-M, and TivTok-L is relatively close. Due to computational resources and efficiency, we used TivTok-S as the reference model for the results in Table 2, and evaluated both TivTok-S and TivTok-L for Table 3.
>
> To address the reviewer’s concern, we have added the corresponding results for TivTok-M and TivTok-L in Tables 2 and 3 in the revised version.
>
> > W6: Analysis of Multiple Time-Invariant (TIV) Tokens.
>
> We thank the reviewer for their interest in this aspect of our work. Using multiple time-invariant tokens naturally tends to **improve reconstruction quality** due to the increased number of tokens. However, this also **reduces the effective compression rate**.
> We provide a detailed analysis of using multiple TIV tokens at 128×256×256 resolution below and in the Appendix. The results confirm the expected trade-off:
>
> - With 4 TIV tokens, reconstruction quality remains higher than the baseline while efficiency is moderately improved.
>
> - With a single TIV token, compression efficiency is maximized (2.91×) while FVD remains comparable.
>
> Overall, this highlights the balance between reconstruction quality and compression rate. **We strongly encourage the reviewer to refer to Figure 6 in the Appendix C for a more intuitive comparison.**
> | Method |TIV Token Numbers | Compression Ratio (%) | T/P (%, tokens-to-pixels) |  PSNR  |  SSIM  |  LPIPS  |  rFVD  | CVVAE |
> |----------|---------|------------------------|----------------------------|--------|--------|---------|--------|--------|
> | CVVAE |-                 | 0.521                  | 0.3906                     | 29.00  | 0.8831 | 0.0729  | 72.91  | —      |
> | TivTok |8          | 0.521                  | 0.0122                     | 30.07  | 0.9003 | 0.0618  | 28.96  | —      |
> | TivTok |4          | 0.326                  | 0.0076                     | 28.97  | 0.8825 | 0.0739  | 39.84  | —      |
> | TivTok |2          | 0.228                  | 0.0053                     | 27.20  | 0.8453 | 0.0951  | 81.18  | —      |
> | TivTok |1          | 0.179                  | 0.0042                     | 26.23  | 0.8210 | 0.1057  | 92.09  | —      |
>
> > W7: Typos.
>
> We thank the reviewer for the careful reading of our paper and for pointing out these typos. We have corrected all of them in the revised version. We sincerely appreciate the reviewer’s attention to detail.

---

### Official Review · Reviewer_Adw3 · 2025-11-03

**Soundness:** 2
**Presentation:** 3
**Contribution:** 2
**Rating:** 2
**Confidence:** 5

**Summary:**

The paper proposes TivTok, a video tokenization method that decouples videos into time-invariant (TIV) tokens and time-variant (TV) tokens to achieve better compression efficiency. The TIV tokens capture shared content across frames, while the TV tokens encode frame-specific details. The approach uses a transformer-based architecture with dual-range attention masks and a TIV Token Broadcasting mechanism to isolate shared versus frame-specific content, facilitating reusability of TIV tokens. The method shows a 2.91× improvement in compression efficiency, while maintaining comparable reconstruction quality

**Strengths:**

Scalability for video length: The proposed method efficiently scales to long videos by reusing TIV tokens across chunks, reducing tokenization complexity from quadratic to linear.

Clear methodology: The transformer-based architecture, attention masking, learned common content and the broadcasting mechanism for token reuse are well-explained.

**Weaknesses:**

Lack of innovation and comparison to previous methods: The core idea of TivTok is very close to the CMD and HiVAE method, which also uses a form of content decomposition. The claim of novelty is unsubstantiated, and the authors do not convincingly distinguish their method from CMD and HiVAE. Although these methods are mentioned and described in the related work section, the direct comparison of these baselines is absent in the experiments.

Meaningless and trivial derivation of Eq. (2)-(4): Equations (2)-(4) provide a trivial proof that adds little value to the paper. The argument can be summarized in one sentence: encoding frames independently consumes significantly more tokens than encoding shared content and frame-specific residuals.

Scalability concerns with TivTok-L: TivTok-L, despite being larger, appears to be inferior to the smaller TivTok-M in short video reconstruction. The related explanation is lacking.

Insufficient ablation studies: The ablation study is too simplistic and lacks critical insights. Specifically, there is no exploration of alternative ways to generate TIV and TV tokens. For example, could using 3D convolutions for generating TIV tokens provide better performance? Furthermore, is there any potential improvement in performance if TV tokens are generated by leveraging adjacent frames, or would this approach merely add computational overhead without significant benefits? The impact of different loss functions is not presented. These aspects should be explored to understand the full impact of different design choices and to validate the effectiveness of the proposed method.

Weak qualitative results: The qualitative results presented in Figure 3 do not demonstrate significant improvements over the baseline. For example, the results in Figure 3(a) for TiVTok are less clear than those from Omni-DV, and the hand reflection on the piano surface in Figure 3(d) is closer to the ground truth in CV-VAE. Furthermore, the description of shoe details in the sumo scene is inaccurate, as the sumo wrestler is not wearing shoes in the competition.

**Questions:**

1. The core idea of TivTok is similar to CMD and HiVAE. Clarify the specific innovations that distinguish TivTok from CMD, and add the comparison in the experiments.

2. Clarify why TivTok-L, despite being larger, appears inferior to TivTok-M in some cases. Explain how the increased computational cost and model size of TivTok-L are justified in terms of performance improvements.

3. Expand the ablation study to include the impact of various design choices, hyperparameters and loss functions.

4. Provide clearer visual examples or more detailed comparisons that highlight TivTok's advantages over existing methods.

---

> ### Author Response · Authors · 2025-11-21
>
> We sincerely thank reviewer Adw3 for the valuable time and for recognizing the scalability and clarity of our methodology. We truly appreciate the thoughtful suggestions, which will further help us improve the quality of our work. Below, we provide our responses to each of the concerns raised.
> > W1 & Q1: Innovation and comparison to previous methods CMD and HiVAE.
>
> We agree that content decomposition is a widely explored idea and that several prior works—including CMD and HiVAE—attempt to factorize videos into different components. However, our contributions differ from these methods in both motivation and methodological design, as detailed below.
> 1. Motivation: Prior works decompose videos into particular factors: CMD aims at content–motion decomposition. HiVAE decomposes videos into high-frequency and low-frequency components for better compression. In contrast, our work pursues a more general objective: We decompose a video into a time-invariant component and a time-variant component, without prescribing each part.
>
> This design has two key implications:
>
>   - More general decomposition. Explicitly defining the factors (e.g., “motion”, “high-frequency”) can be restrictive and may not generalize to longer sequences or more complex video structures. Our formulation avoids this limitation.
>
>   - Enables reusability of the time-invariant component (TIV token). The time-invariant representation can be reused across all time steps during long video tokenization.
>
> These conceptual distinctions were also recognized by reviewers:
> >“I like the idea of decomposing videos into time-invariant parts and time-variant parts. There have been similar approaches, but I think this approach tries to do a more explicit decomposition than prior works, as well as showing better reconstruction quality.” — Reviewer F2Sa
>
> >“I think the idea is very clever. Redundancy has generally been limited to pixel-level (spatial / temporal redundancy) rather than semantic, and the TIV tokens are a nice way to incorporate this. ” — Reviewer VFj8
>
> 2. Methodological design:  Our method introduces a single transformer-based tokenizer that jointly encodes both components into one 1D latent sequence, enabled by two new mechanisms: A. Dual-range encoder via a customized attention mask guides different tokens to capture time-invariant vs. time-variant information. B. Time-invariant token broadcasting in the decoder allows the time-invariant representation to be reused across all time steps. These designs make our method structurally distinct from CMD and HiVAE. CMD decomposes videos into 2D content frames and 1D motion representations using two separate models, and HiVAE similarly splits videos into high- and low-frequency components with two independent encoders. Because their two components lack a unified latent space and effective cross-interaction, redundant information cannot be fully compressed.
>
> These method design were also recognized by reviewers:
> >“I think the mechanism of using the invariant tokens and keeping time-variant tokens separate per frame is a nice way to parallelize the decoding and potentially decrease the memory and time usage while also making generation potentially faster. ” — Reviewer VFj8
>
> The direct comparison of these baselines is absent in the main experiments because of differences in experimental settings. Nevertheless, we have conducted the comparison between CMD and HiVAE, as shown below, and we have also included the results in Table 5 of the revised paper.
>
> | Method          | Dataset | Compression Ratio |   PSNR  |   SSIM   |  LPIPS  |  FVD   |
> |-----------------|---------|-------------------|---------|----------|---------|--------|
> | CMD             | UCF-101 | 10.42%            |   -     |    -     |    -    |  7.72  |
> | CMD-reproduced  | UCF-101 | 10.42%            | 30.12   | 0.9201   | 0.0243  | 81.91  |
> | Ours            | UCF-101 | 8.33%             | 32.64   | 0.9533   | 0.0123  |  7.14  |
> | CMD             | WebVid  | 6.85%             | 26.553  | 0.795    | 0.11    | 98.623 |
> | HiVAE           | WebVid  | 0.27%             | 29.347  | 0.834    | 0.096   | 61.941 |
> | Ours            | WebVid  | 0.26%             | 28.61   | 0.8288   | 0.0729  | 22.96  |
> | Ours            | WebVid  | 0.52%             | 31.69   | 0.8958   | 0.0477  |  7.15  |
>
> As shown in the results, our method achieves better reconstruction quality across all metrics under the same or even higher compression ratios, owing to our more general decomposition strategy. Note that CMD does not provide PSNR, SSIM, LPIPS metrics in its paper; therefore, we reproduced its results using the official repository. However, the official repo includes the following disclaimer:‘It's possible that this code may not accurately replicate the results outlined in the paper due to potential human errors during the preparation and cleaning of the code for release.’  For completeness and transparency, we report both the reproduced  and the reported results here.

---

> ### Author Response · Authors · 2025-11-21
>
> > W2: Revision of the Entropy Discussion.
>
> We thank the reviewer for pointing this out. Equations (2)-(4) were intended to provide a systematic justification of the motivation. Although the conclusion — that encoding frames independently consumes significantly more tokens than encoding shared content and frame-specific residuals — may be obvious to experienced readers, we have simplified this part in the main paper and moved the full derivation to the Appendix A. This keeps the paper concise while still providing a complete explanation for interested readers.
>
> > W3 & Q2: Scalability concerns.
>
> We apologize for the omission of details in the paper and thank the reviewer for pointing this out. **TivTok-S, TivTok-M, and TivTok-L have the same overall model size, and the only difference lies in the trade-off between the number of tokens and the token dimensionality.** As shown in Table 1, moving from S → M → L increases the number of tokens while maintaining the same absolute compression rate. To further address the reviewer’s concern, we also provide a scalability experiment below and include the results in Table 5 of the revised paper. The experiment shows that our method benefits from larger model configurations and achieves progressively better performance as the model size increases.
>
> | Model Configuration |  PSNR  |   SSIM   |  LPIPS  |  FVD  |
> |---------------------|--------|----------|---------|-------|
> | ViT-Small           | 27.73  | 0.8611   | 0.0809  | 47.31 |
> | ViT-Base            | 30.13  | 0.9010   | 0.0614  | 21.29 |
> | ViT-Large           | 30.94  | 0.9116   | 0.0490  | 13.11 |

---

> ### Author Response · Authors · 2025-11-21
>
> > W4 & Q3: More ablation study.
>
> We conducted a comprehensive ablation study in the paper (Table 4) that evaluates all components specifically designed for our framework, including (1) the core time-invariant/time-variant decomposition, (2) the dual-range encoder with the tailored attention mask, (3) TIV-token reuse and the parallel-decoding architecture, and (4) the proposed TIV-Broadcasting training strategy. These studies collectively demonstrate that each component is essential for effectively extracting and reusing video redundancy.
> We thank the reviewer’s interest in our method design and are glad to elaborate on and address our considerations regarding the points raised.
>
> 1. Design choices: Why not use 3D convolutions for generating TIV tokens?
>
> The purpose of TIV tokens is to extract and reuse information that is invariant across time. This requires:
> - Global temporal visibility, so the TIV representation can observe all frames simultaneously;
> - A clean separation of information flow between time-invariant and time-variant features, so that TIV tokens encode only invariant content and TV tokens encode the complementary varying content.
>
> Transformers—with their global receptive field and the ability to enforce visibility constraints via attention masks—naturally satisfy these requirements. In contrast, 3D convolutions are inherently local in both space and time. They cannot observe all frames at once, nor can they implement hard visibility constraints needed to enforce the decomposition. Therefore, 3D convolutions cannot serve as a drop-in mechanism for extracting TIV tokens, as TIVs are defined not by spatial/temporal tensor shape but by their information semantics and controlled interaction pattern.
>
> 2. Leveraging adjacent frames for generating TV tokens
> Table 4 already provides an ablation of the dual-range encoder in which TV tokens are extracted by interacting with all frames. Notably, allowing TV tokens to see all frames leads to degraded performance, particularly in long-video tokenization.
> This occurs because unconstrained visibility leads to information leakage:
> - Some time-variant information is absorbed by TIV tokens;
> - Some time-invariant information gets encoded into TV tokens.
>
> When reusing TIV tokens across long sequences, misallocated information results in noticeable performance drops. To directly address the reviewer’s suggestion, we additionally conducted an experiment where TV tokens are extracted using adjacent frames. The results further confirm that our dual-range encoder with the proposed attention mask is necessary to maintain a clean decomposition and stable performance.
> | TV Token Source  |  PSNR  |   SSIM   |  LPIPS  |   FVD    |
> |------------------|--------|----------|---------|----------|
> | Single Frame     | 29.05  | 0.8831   | 0.0719  |  38.49   |
> | Adjacent Frame   | 26.77  | 0.8421   | 0.0943  | 104.21   |
> | All Frame        | 19.67  | 0.5691   | 0.3525  | 1359.38  |
>
> 3. Impact of different loss functions
> Our method does not introduce any special loss designs. We follow the training objectives commonly adopted in prior video tokenization work [1–5], which is consistent with the broader literature. For completeness, we summarize the effects of different loss functions in the Appendix and below, as these behaviors have been consistently reported in prior literature and may be helpful for readers interested in this direction.
>
> - L1 loss encourages accurate pixel-level reconstruction and is the most stable term for training tokenizers.
>
> - Perceptual loss improves texture sharpness and semantic alignment by comparing features in a pretrained network, mitigating over-smoothed outputs.
>
> - Adversarial loss enhances realism and high-frequency details, though it typically contributes less to overall bitrate–quality trade-offs than the reconstruction loss.
>
> [1] Chen, H., Wang, Z., Li, X., Sun, X., Chen, F., Liu, J., ... & Barsoum, E. (2025). Softvq-vae: Efficient 1-dimensional continuous tokenizer. In Proceedings of the Computer Vision and Pattern Recognition Conference (pp. 28358-28370).
>
> [2] Wang, H., Suri, S., Ren, Y., Chen, H., & Shrivastava, A. LARP: Tokenizing Videos with a Learned Autoregressive Generative Prior. In The Thirteenth International Conference on Learning Representations.
>
> [3] Wang, Y., Guo, J., Xie, X., He, T., Sun, X., & Bian, J. (2025). Vidtwin: Video vae with decoupled structure and dynamics. In Proceedings of the Computer Vision and Pattern Recognition Conference (pp. 22922-22932).
>
> [4] Zhao, S., Zhang, Y., Cun, X., Yang, S., Niu, M., Li, X., ... & Shan, Y. (2024). Cv-vae: A compatible video vae for latent generative video models. Advances in Neural Information Processing Systems, 37, 12847-12871.
>
> [5] Wang, J., Jiang, Y., Yuan, Z., Peng, B., Wu, Z., & Jiang, Y. G. (2024). Omnitokenizer: A joint image-video tokenizer for visual generation. Advances in Neural Information Processing Systems, 37, 28281-28295.

---

> ### Author Response · Authors · 2025-11-21
>
> >W5 & Q4: Qualitative results concerns.
>
> We sincerely thank the reviewer for their careful examination and valuable feedback. We would like to clarify that in Figure 3, we compare 128×256×256 video reconstructions. For reference, CV-VAE achieves a compression rate of 0.521%, while Omni and Omni-DV achieve 1.04%. **By reusing TIV tokens, our method attains a much lower compression rate of only 0.179%. Given this substantial difference, we did not expect “significant improvements over the baseline” in visual fidelity.**
>
> The goal of Figure 3 is to demonstrate that, despite this much lower compression rate, our method still maintains good visual quality and preserves certain details better than the baselines. **To further support this, we have included additional qualitative results in Figure 9 of the Appendix, showing that even with a significantly lower compression rate, our approach achieves reconstruction quality comparable to the baselines.**

---

### Author Response · Authors · 2025-12-03
**Summary of Rebuttal and Discussions**

Dear Reviewers, ACs, SACs, and PCs,

Thank you for your time and effort in reviewing our work. We recognize that the workload is especially heavy this year. To help streamline the discussion, we provide a summary below. **We hope this will assist the ACs in quickly navigating the key issues addressed in our rebuttal and make the overall discussion easier to follow.**

---

## Strength Summary

We first sincerely thank the reviewers for their recognition of our work:

- **Novel and Well-Motivated Idea:** Reviewers found the idea *“a more explicit decomposition into time-invariant and time-variant parts”* (**F2Sa**), *“very clever”* in using semantic redundancy via TIV tokens (**VFj8**), and *“solid and technically sound”* in motivation (**hY2j**).

- **Clear and Technically Sound Methodology:** The methodology was described as *“clear”* with well-explained components (**Adw3**), *“technically sound”* (**hY2j**), and effective for parallel decoding via invariant/variant token separation (**VFj8**).

- **Promising Results:** The approach achieves *scalability to long videos* (**Adw3**), *decent reconstruction with lower computation budget* (**hY2j**), and *a great compression ratio* compared with existing methods (**F2Sa**).

- **Clear and High-Quality Presentation:** Reviewers repeatedly described the paper as *“clearly written and easy to follow”* (**hY2j**, **F2Sa**, **VFj8**).

---

## Concerns and Our Response

- **Comparison with existing content-decomposition methods CMD (ICLR’24) and HiVAE (arXiv’26) (Adw3):**
  Our approach differs fundamentally from CMD and HiVAE in both idea and method. As highlighted by reviewers **F2Sa** and **VFj8**, our decomposition strategy is more explicit and clever. Moreover, our design introduces the **novel idea of reusing time-invariant tokens**, which—to the best of our knowledge—is the first to exploit redundancy for effective long-video compression. Methodologically, unlike CMD and HiVAE, we use an **attention mask** to unify invariant and variant latents in a single 1D token space, yielding a simpler and more scalable design. We also include direct comparisons with CMD and HiVAE in this rebuttal and **Table 5**, which **experimentally demonstrate** the effectiveness of our method.

- **Scalability concerns in model size (Adw3, hY2j) and dataset size/resolution (F2Sa):**
  We supplemented experiments with different model sizes, dataset sizes, and resolutions in **Table 5**. The results **experimentally demonstrate** that our method consistently improves as the model and dataset size increase, and can effectively handle higher-resolution videos, highlighting the **strong scalability** of our approach.

- **Further analysis on TIV and TV tokens (hY2j, F2Sa, VFj8):**
  - *Systematic analysis:* Experiments in **Table 8/Figure 6** (TIV token number, **hY2j**) and **Table 9/Figure 7** (TIV/TV allocation, **VFj8**) show **2.51× smaller FID** at the same compression ratio and comparable performance at **2.91× higher compression**, demonstrating flexibility and effectiveness.
  - *Decomposition validation:* **Figure 8** (**hY2j, F2Sa**) shows that fixing TIV tokens while varying TV tokens generates different videos, confirming the decomposition and token reuse. Additional analysis (**VFj8**) illustrates that TIV tokens’ **global view captures highly redundant information**, enabling effective long-video tokenization and compression.

- **Additional generation experiments and performance metrics (F2Sa, VFj8):**
  Expanded experiments in **Table 3** include more long-video results, additional baselines (**F2Sa**), and **Wall-clock time and memory** measurements (**VFj8**), demonstrating the **effectiveness and efficiency** of our method.

- **Writing improvements (Adw3, hY2j, VFj8):**
  Simplified the entropy discussion and moved details to the supplementary material (**Adw3, VFj8**), provided more details in Table 6/7, corrected typos (**hY2j**), improving the **clarity, readability, and overall presentation**.

---

## Summary of Discussion

- Due to this year’s special circumstances, we were only able to have a thorough discussion with reviewer **F2Sa**, who indicated that their concerns have been addressed and raised the score from 6 to 8. We sincerely appreciate their support, citing comments such as *“I think the paper is strengthened more, and now I think this is a quite good paper”* and *“I want to champion this paper and raise the score accordingly!”*

- We also observed positive attitudes from other reviewers (**hY2j, VFj8**), who expressed willingness to increase their scores.

- Throughout, we have made every effort to address all concerns and are grateful for the valuable feedback that helped further improve our paper.

Finally, we thank the AC and reviewers again for their constructive contributions to improving this paper. We hope this summary assists in the final assessment of our work.

Sincerely,

ICLR 2026 Submission 2988 Authors

---

### Meta-Review · Area_Chair_s9SD · 2026-01-04

**Summary:**

The reviewers identified several critical issues that informed the decision for rejection:

* Limited algorithmic novelty: The core idea of "time-invariant/variant decomposition" is heavily grounded in prior works like CMD and HiVAE, as well as the image-based TiTok. Reviewers felt the jump to the current architecture was primarily an orchestration of existing ideas.

* Weak qualitative fidelity: Despite gains in compression efficiency (2.91×), the reconstruction quality was cited as unconvincing for a generative modeling bottleneck, with specific artifacts noted in the visual examples.

* Incremental theoretical insight: The entropy-based motivation (Eqn. 2–4) was viewed as a "trivial proof" that added little value to the research beyond stating that shared content is more efficient to encode than independent frames.

* Scalability inconsistencies: The larger "TivTok-L" model performed worse than the mid-sized "TivTok-M" in several short-video reconstruction cases, raising doubts about the robustness of the architectural trade-offs between token count and dimensionality.

**Reviewer Concerns:**

Concerns Addressed by the Rebuttal

* Baseline comparisons (`Adw3`, `F2Sa`): The authors provided new comparative experiments against CMD and HiVAE. On the UCF-101 dataset, TivTok achieved a PSNR of 32.64 vs. CMD's 30.12 at similar compression ratios.

* Efficiency metrics (`VFj8`): The authors added wall-clock time and GPU peak memory measurements. They showed that TivTok reduced total decoding time for a 128-frame video from 2.14s (CVVAE) to 0.21s.

* Scalability study (`hY2j`, `F2Sa`): Results on larger datasets like VidProM were added to Table 5, showing that the model maintains stability at higher resolutions (512p).

Outstanding Concerns

* Unsubstantiated novelty claim (`Adw3`): This remains the primary blocker. Reviewer Adw3 noted:
> _"The core idea of TivTok is very close to the CMD and HiVAE method... The authors do not convincingly distinguish their method from CMD and HiVAE."_

  While the authors argued that their decomposition is "more general," the underlying strategy of factorizing static vs. dynamic components remains a well-trodden path in video compression research.

* Redundancy between tokens (`hY2j`): The reviewer questioned if TV tokens truly encode "only" frame-local residuals. The authors admitted in the rebuttal:
> _"Achieving perfect disentanglement is inherently challenging... certain information may still appear in the TV tokens."_

  This undermines the key claim of the paper that the model learns a clean, time-invariant semantic space.

* Qualitative quality vs. compression trade-off (`Adw3`): Reviewer `Adw3` found the Sumo wrestler scene and piano reflections to be less clear than baselines. The authors defended this by citing the lower compression rate, but for a conference focused on Generative Models, the loss of fidelity is a significant drawback.

**Reviewer Scores:**

* Reviewer `Adw3` (Initial: 2 $\rightarrow$ Estimated: 3): While the empirical comparisons were provided, the reviewer’s core critique was about conceptual overlap with prior work. These are fundamental disagreements that are rarely resolved by new numbers alone.

* Reviewer `hY2j` (Initial: 4 $\rightarrow$ Estimated: 4): The author's concession that "perfect disentanglement is theoretically difficult" likely kept this reviewer concerned about the efficiency of the token representation.

* Reviewer `VFj8` (Initial: 6 $\rightarrow$ Estimated: 6): The addition of wall-clock time addressed their primary request, but they did not act as a strong champion for acceptance against the novelty critiques.

* Reviewer F2Sa (Initial: 6 $\rightarrow$ Estimated: 7): The reviewer expressed high enthusiasm after the rebuttal, with the authors reporting an intended upgrade to a score of 8. In light of this cycle's special circumstances and the interrupted AC-reviewer discussion phase, this meta-reviewer is estimating the final sentiment more conservatively. While the reviewer intended to "champion" the work, their perspective was not fully reconciled with the fundamental novelty concerns raised by Reviewer `Adw3`.

---

### Decision · Program_Chairs · 2026-01-26

Reject